



# Surrogate model-based precipitation tuning for CAM5

Xianwei Wu[1], Liang Hu[1], Lanning Wang[2], Haitian Lu[4], and Juepeng Zheng[3]

[1]School of Computer Science and Technology, Jilin University, Changchun, China
[2]College of Global Change and Earth System Science, Beijing Normal University, Beijing, China
[3]School of Artificial Intelligence, Sun Yat-Sen University, Zhuhai, China
[4]National Supercomputing Center in Wuxi, Wuxi, China

**Correspondence:** Juepeng Zheng (zhengjp8@mail.sysu.edu.cn)

**Abstract.** The uncertainty of physical parameters is a major reason for a poor precipitation simulation performance in Earth system models (ESMs), especially over the tropical and Pacific regions. Although tuning related parameters can help reduce such uncertainty factors, repetitive runs of ESMs incur large computational costs. While surrogate models can reduce the computational costs in many tuning scenarios, building an effective surrogate model for the community atmospheric model (CAM) is a complex integration of many processes, which is an unresolved challenge due to its strong nonlinear behaviors. In

this study, we present a surrogate model-based parameter tuning framework for the CAM and apply it to improve the CAM5 precipitation performance. We propose a multilevel surrogate model-based optimization method. First, a global-level surrogate model is constructed with a gradient boosting regression tree (GBRT), which has been proven, through cross-validation experiments, to have a more significant effect than other methods. The candidate point approach (CAND) is applied to bal-

ance exploration and exploitation to obtain better values for establishing a local-level surrogate model. A local-level surrogate model is then constructed based on a significantly reduced number of selected points.. We design a trust region approach to adjust the sampling region during the tuning process. This proposed method has a faster convergence speed and higher accuracy during the tuning process. We attempt a region-based optimization method to improve the CAM simulation results over some areas with large errors. The results show that the surrogate model-based optimization method can significantly improve

the simulation performance of the CAM model. The average improvement of the selected regions is 19%. To integrate the optimization results of these regions, we design a nonuniform parameter parameterization scheme and integrate the parameters using a parameter smoothing scheme, and the experimental results improve in four regions. These simulation experiment results of CAM5 demonstrate that this proposed surrogate model-based method improves the precipitation simulation of the CAM model.

## 1 Introduction

The Earth system model (ESM) is an indispensable tool for predicting future climate change trends. In ESMs, because of the grid resolution constraint, physical parameterization schemes are used to describe subgrid physical processes. These subgrid-scale parameterizations involve interactions with continental hydrology, aerosol microphysics, radiation, thermodynamics, chemistry and biology (Hourdin et al., 2017). Each of them contains many uncertain parameters, which represent global



processes. These parameters control the physical processes at the subgrid scale, and slight variations could lead to huge devi-
ations in the simulations. Therefore, it is important to calibrate the parameters to better capture real world physical behaviors.
Generally, parameter tuning depends on the experience of climate model experts (Wu et al., 2020). However, the physical pro-
cesses in the Earth system model are becoming increasingly complex as atmospheric science continues to advance. Traditional
expertise-based tuning methods have gradually become less useful.

Automatic optimization algorithms are effective tools to replace the manual methods. There have been many studies that used
various optimization techniques to achieve parameter tuning. For example, Yang et al. (2012) used multiple very fast simulated
annealing (MVFSA) (Ingber, 1989) to tune the Kain-Fritsch (KF) convective parameterization and improved weather research
and forecasting (WRF) simulation performances over the Southern Great Plains (SGP) region. Afshar et al. (2020) attempted
to tune the parameters of the Kain-Fritsch scheme in the WRF model to reduce the uncertainty in quantitative precipitation
prediction. These models, with relatively simple calculation processes, can find better parameters through a certain number of
iterations.

When dealing with parameter tuning of complex ESMs such as CAM, the abovementioned methods may become infeasible.
Because these models require a long-time simulation to achieve a steady state, the computational cost incurred by the iterations
of the optimization method is unacceptable. Effectively reducing the optimization time is an important problem to be considered
for the parameter tuning methods for these complex ESMs. Many studies focus on solving the problem; for example, Yang et al.
(2013) proposed a simulated stochastic approximation annealing (SSAA) (Liang et al., 2014)-based parameter tuning method.
The parameters of the Zhang-McFarlane scheme in CAM5.3 were tuned by the SSAA, and there were some positive impacts,
including a double intertropical convergence zone and East Asian monsoon precipitation after using the optimal parameters.
To improve the optimization efficiency, Zhang et al. (2015) enhanced the downhill simplex optimization technique, leading
to the discovery of a better local minimum solution by selecting more appropriate initial parameter values. Simulation results
on the grid-point atmospheric model of IAP LASG version 2 (GAMIL2), in limited optimal iterations, proved the downhill
simplex method was better than the global optimization algorithms. Zhang et al. (2018) proposed an automated tuning approach
that integrates automatic tuning with short-term hindcasts to alleviate the heavy computational workload. The tuning led to
a substantial reduction in the significant underestimation of the CAM5 longwave cloud forcing, and the overestimation of
precipitation. These studies indicated that such algorithms can calibrate parameters automatically, and effectively improve the
simulation accuracy of the ESMs. However, the quality of the solution is related to the number of iterations, and a better
solution implies more iterations. Typically, ESMs are more complex and resource-consuming than other problems. Therefore,
the computational cost generated by such iteration processes is usually unacceptable. Although these methods shorten the
optimization time to varying degrees, compared to the traditional parameter tuning methods, they still need to run an ESM
many times.

Surrogate models are regarded as effective tools to solve computationally expensive optimization problems (Sun et al.,
2017). The objective function of these problems is difficult to describe as a mathematical expression; the mathematical ex-
pression is complex and time-consuming. The key to solving these problems is to reduce the number of expensive model
runs to an acceptable level. A surrogate model establishes connections between adjustable parameters and the responses of a





complex expensive model by statistical or data-driven models to approximate a complex expensive model. This is one of the most commonly used approaches to optimizing large complex models. Recently, some studies have indicated that there are various methods to construct surrogate models, such as polynomial regression (Lian and Liou, 2005), support vector machines (Vapnik, 1999; Loshchilov et al., 2010), radial basis function (RBF) networks (Gutmann, 2001) ANNs (Gaspar-Cunha and Vieira, 2005) and Gaussian processes (GPs, alternatively known as Kriging or models for design and analysis of computer

experiments (DACE)) (Rasmussen, 2003; Jie et al., 2017). Based on these studies, some optimization methods were proposed for parameter optimization. For example, efficient global optimization (EGO) (Jones et al., 1998) used the Kriging method. Dynamic coordinate search using response surface models (DYCORS) (Regis and Shoemaker, 2013) improved the RBF surrogate model for high-dimensional expensive models. However, most of these surrogate model optimization methods focused on mathematical function benchmarks. It is obvious that the optimization for the mathematical functions can differ significantly

from optimizing parameters in complex real-world models.

In recent years, with the development of surrogate model-based optimizations, these methods have been widely used to solve the parameter tuning problems of complex geoscientific models. Neelin et al. (2010) constructed a polynomial model that can approximate an atmospheric general circulation model (AGCM) to guide parameter choices. Müller et al. (2015) used the RBF surrogate model to optimize the methane emission predictions of the community land model (CLM). Xu et al. (2018)

proposed a surrogate model-based optimization method to improve the prediction accuracy of the soil organic carbon (SOC) results applying the calibrated parameter values, and result shows that the error could be reduced by up to 12%. Wang et al. (2014) established a connection between the optimization of mathematical benchmarks and complex geoscientific models, and proposed the adaptive surrogate model-based optimization (ASMO) method; a SCA-SMA hydrologic model was tuned based on the ASMO method. On this basis, ASMO was used to calibrate the parameters of various complex geoscientific models.

For example, Gong et al. (2015) focused on the Heihe river basin and attempted to tune the parameters which belong to the common land model (CoLM) by a MO-ASMO method. Di et al. (2018) implemented the ASMO method to tune WRF model parameters to increase summer precipitation simulations over the Greater Beijing area. Chinta and Balaji (2020) presented a MO-ASMO method to enhance the prediction accuracy of the Indian summer monsoon (ISM) in the WRF model. Zhang et al. (2020) proposed a land surface evapotranspiration (ET) optimization method based on the multivariate adaptive regression

splines (MARS) surrogate model and the ASMO method. The mentioned studies have sufficiently shown that surrogate model-based optimization methods are capable of resolving the ESM parameter tuning problems. However, most of these studies focused on weather models or land models, such as WRF, CoLM and CLM, which cannot prove that the surrogate model-based optimization methods are effective for CAM parameter tuning. CAM is strongly nonlinear and contains more complex physical processes. The application of surrogate models for CAM requires further exploration. It is well known that the CAM model

is a complex ESM, and CAM5 cleanly separates the parameterization suite from the dynamical core(Neale et al., 2010). The precipitation process is complicated (Dai, 2006), and the precipitation-related processes include cumulus convection processes, large-scale circulation, microphysical processes, boundary layer processes, etc. Zhang and McFarlane (1995); Morrison and Gettelman (2008) showed that the precipitation process is related to several physical quantities. In addition, the intensity of the precipitation and the influence of the parameters show a strong nonlinear relationship, which is hard to describe through





a mathematical function. Anderson and Lucas (2018) used the random forest method to predict global mean precipitation; however, it only established precipitation predictions based on global average values and could not resolve the precipitation errors based on the precipitation distributions. It is challenging to construct an effective surrogate model for CAM precipitation and tune the parameters based on a surrogate model.

The CAM is a well-calibrated model (Zhang et al., 2018). Many simulation experiments have indicated that CAM5 captures
the global scale of the precipitation characteristics reasonably well (Qian et al., 2015; Chen and Dai, 2019). However, the simulation results are still not accurate in some areas. Improving the simulation results over the areas with errors has become one of the biggest CAM simulation challenges, and many studies have attempted to improve the CAM simulation results over these error areas. Li et al. (2015) analyzed the precipitation climatological features over East Asia. Pathak et al. (2021) attempted to improve a CAM5 precipitation simulation over South Asia. Wang et al. (2016) improved the tropical precipitation
variability simulation results in CAM5. These studies demonstrated, that compared with the default experiment, the simulation results over some regions can be improved to a certain degree. In this study, a region-based parameter tuning experiment is proposed, and the results show that the results over each region are better than those of the default experiment.

The main research contributions of this work are as follows:

1. We first propose a surrogate model-based parameter tuning method for CAM5 precipitation in this study. Considering
that the nonlinearity and complexity of CAM are much higher than those of other models, such as WRF, the effectiveness and feasibility of the method are validated in this work.

2. We design a multilevel surrogate model method. The multilevel surrogate model method integrates the CAND approach and trust region to update the surrogate model in each iteration. The results show that the proposed method has a faster convergence speed and fewer errors during the tuning process.

3. We design a nonuniform parameter parameterization scheme, and integrate parameters using a parameter smoothing scheme. We explore the influence of the same parameter on precipitation over different regions. Then, a region-based optimization is proposed based on this result, and we construct different surrogate models for each area. The average improvement of the selected regions is 19%. The nonuniform parameter parameterization scheme attempts simultaneous optimizations for as many regions as possible, and the experimental results are improved in four regions.

The structure of this paper is as follows. In section 2 we provides the details of the experimental design and the description of the model. Section 3 introduces the surrogate model-based parameter optimization method. In Section 4, we prove the robustness and effectiveness of the surrogate model construction, and provide evaluations and analyses of the optimization results. Section 5 comprises the conclusion and discussion.

## 2   Design of experiment

### 2.1   Model description

In this study we select the community atmospheric model (CAM Version 5.3), which serves as the atmospheric component of the community earth system model (CESM, Version 1.3). The model adopts the spectral element dynamical core (SE-





dycore) formulation. The CAM is integrated with a data ocean model (DOCN), an active community land model (CLM) and a thermodynamics-only sea ice model (CICE).

## 2.2 Experimental design

**Table 1.** Regions selected in this study

| Name | Region |
|---|---|
| WarmPool | -15°-15°N, 120°-150°E |
| South Pacific | -30°-0°N, 190°-250°E |
| Niño | 6°-12°N, 210°-270°E |
| South America | -20°-0°N, 280°-300°E |
| South Asia | 0°-10°N, 75°-115°E |
| East Asia | 15°-40°N, 105°-140°E |

A series of AGCM simulations are carried out for 6 years (with 1 year as the model spin-up) for all the simulations. The compset used in this study is F_2000_CAM5, and the resolution is ne30_g16. We use the climatological SST dataset for DOCN input data.

Precipitation plays a crucial role as a key physical process. In ESMs, numerous uncertain parameters related to the representations of cloud microphysical processes and deep and shallow convection scheme have a significant impact on precipitation (Qian et al., 2015). Therefore, the parameters we considered included the ZM deep convection scheme(Zhang and McFarlane, 1995), the UW shallow convection scheme(Park and Bretherton, 2009) and the cloud fraction(Morrison and Gettelman, 2008). Considering the fact that the model parameters have significant regional dependence, we split the globe into six main regions to represent the key features of the whole global variable (i.e., the total precipitation rate, PRECT) as (Yang et al., 2013) to conduct the surrogate model simulations, which include, the WarmPool, South Pacific, Niño, South America, South Asia and East Asia regions. We construct 6 surrogate models for each region, and the optimization process described in Section 3 is implemented 6 times. Our results show 5-year climate runs, which fulfill our need to look at the tropics. In addition, we compare the results of the parameterization of the surrogate model with the ERA5(Hersbach et al., 2020) precipitation rate to verify our optimization effect.

The parameter ranges and default values are shown in Table 2. Previous studies have identified these parameters as being sensitive to precipitation. Qian et al. (2015) and Pathak et al. (2020) indicated that the threshold relative humidity for the stratiform low clouds (cldfrc_rhminl) Makes the most significant contribution to the variance of both global ocean mean precipitation and global mean precipitation. The parameter named zmconv_tau which representes the time scale for the consumption rate of deep convective available potential energy (CAPE) is regarded as an important parameter for global total precipitation and deep convective precipitation (Yang et al., 2013). Pathak et al. (2020) showed that the autoconversion size threshold for the ice to snow (micro_mg_dcs) has a significant impact on the variance in the large-scale precipitation rate, convective precipitation





**Table 2.** CAM5 parameter descriptions, default values and ranges. The CAPE represents the convective available potential energy.

| Parameter | Description | Range | Default |
|---|---|---|---|
| cldfrc_rhminl | Threshold relative humidity for the stratiform low clouds | $0.80 \sim 0.99$ | $0.8975$ |
| zmconv_dmpdz | Parcel fractional mass entrainment rate | $-2.0 \times 10^{-3} \sim -0.2 \times 10^{-3}$ | $-1.0 \times 10^{-3}$ |
| zmconv_c0_ocn | Deep convection precipitation efficiency over the ocean | $1.0 \times 10^{-3} \sim 0.1$ | $0.045$ |
| zmconv_tau | Time scale for consumption rate deep CAPE | $1800 \sim 28800$ | $3600$ |
| micro_mg_dcs | Auto conversion size threshold for the ice to snow | $100 \times 10^{-6} \sim 500 \times 10^{-6}$ | $400 \times 10^{-6}$ |
| micro_mg_ai | Fall speed parameter for the cloud ice | $300 \sim 1400$ | $700$ |

rate and total (convective + large-scale) precipitation rate. The parcel fractional mass entrainment rate (zmconv_dmpdz) is a parameter with high influence on both the global mean precipitation and the total variance of extremes precipitation. However, its contribution to the variance of global land mean precipitation or global ocean mean precipitation is relatively smaller. In sensitivity experiments related to precipitation, the fall speed parameter for the cloud ice (micro_mg_ai) has been recognized as a parameter with significant impact (Sanderson et al., 2008). In CAM5, the deep convection precipitation efficiency over the ocean (zmconv_c0_ocn) is exclusively applied in oceanic regions. Nevertheless, it also has a notable impact on the precipitation variance over certain land areas, demonstrating substantial nonlocal effects and feedback on land precipitation from processes occurring over the ocean (Qian et al., 2015).

## 3 Method

### 3.1 The surrogate model-based tuning method procedure

The surrogate model-based parameter tuning method involves several steps. To begin with, the initial sample sets of these parameters are created through a sampling method. Then, these sample sets are utilized as input parameters to conduct the CAM model simulation and to calculate the RMSE objective function value. Second, the global-level surrogate model is created by matching these samples and the corresponding CAM simulation results. A surrogate model establishes connections between the adjustable parameters and the responses of a complex expensive model by statistical or data-driven models to approximate the complex expensive model. In each iteration, a distinct strategy is employed to generate new sample points, and the points are treated as input parameters to execute the real model. The strategy leverages the information and knowledge obtained from the surrogate model to optimize the run time of the real complex model to fulfill the requirement of accuracy. The recently generated sample points and their corresponding simulation outputs are added to the initial sample sets and are used to update the global-level surrogate model until global-model convergence. Then, a local-level surrogate is constructed using a significantly smaller number of sampling points with high-quality CAM model simulation results, to avoid falling into a local optimum. Furthermore, strategies for dynamically changing the search space are applied to update the local-level surrogate model. Finally, once the convergence criteria of the local-level surrogate model are met, the tuning method finishes and outputs





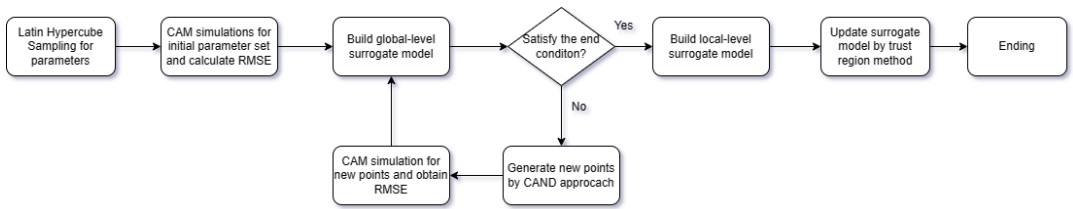

**Figure 1.** The flowchart of the multilevel surrogate model-based parameter tuning method

the optimized parameter values. In the parameter tuning process, each surrogate model can fully explore the parameter space to obtain better solutions, generating a large number of samples. Just a limited number of selected promising parameter points will be sent to the real complex model. By avoiding simulations with low-quality parameters, we can significantly reduce the quantity of meaningless real model simulations. Therefore, the surrogate replaces the actual complex model, and the computation cost will be substantially reduced during the tuning process. A specific implementation showing how to tune

CAM5 precipitation parameters using this surrogate model-based method is illustrated in Figure 1 and is described in Algorithm 1.

---

**Algorithm 1** The multilevel surrogate model-based tuning method for CAM precipitation

---

1: Generate sampling set by Latin hypercube sampling method.

2: Run the CAM model with the parameter set in Line 1 and calculate corresponding RMSE.

3: Construct GBRT global-level surrogate model based on the sampling set.

4: **while** End condition is not met **do**

5:     Selected the next point as input to run CAM model by CAND strategy.

6:     Run CAM model with new parameter in Line 5 and calculate the RMSE of the new point.

7:     Update the surrogate model using new parameter and RMSE value in Line 6.

8: **end while**

9: Construct GP local-level surrogate model using a few high-quality sampling points results.

10: Update the local-level surrogate model and trust region.

11: **return** The the tuning results of CAM precipitation.

---

## 3.2 Global-level surrogate model

### 3.2.1 Initial Sampling

For the initial sampling, the sampling method affects the accuracy of the surrogate model. Monte Carlo sampling (Sobol', 1967;

Halton, 1964) and Latin hypercube sampling (LHS) (Mckay and Conover, 1979) are appropriate methods for constructing the surrogate model because of their flexible sample sizes and good space filling capabilities, and with relatively few points, they cover the entire sample space (Wang et al., 2014). However, the values used in Monte Carlo sampling are randomly drawn





based on a probability distribution, and because of the complexity of CAM, it is difficult to attain additional knowledge of probability distributions about the CAM model simulation results and corresponding parameters. Therefore, we choose the

LHS sampling method as the initial sampling method. It is a stratified sampling method. In contrast to random sampling, this method maximizes the stratification of each edge distribution so that guarantees complete coverage of each variable range. In contrast to random sampling, LHS guarantees that the collection of random numbers accurately represents the true variability of these parameters (Iman et al., 1981).

In addition to the sampling method, the number of samples is also a key factor. Regis and Shoemaker (2007) attempted to set

the number of samples equal to $2(d+1)$, and $d$ is the number of dimensions of the problem; however, CAM is more complex and more nonlinear, and too few initial samples will seriously affect the convergence speed. Gong et al. (2015) and Wang et al. (2014) analyzed the relationship between the performance of the surrogate model and the quantity of samples. When the number of samples exceeds 20 times the number of parameters, the performance of the surrogate model will not improve with an increase in the number of samples. Therefore, considering the efficiency of the method and parameters calibrated in this

study, the initial sample set is defined as 60, each sample consists of 6 parameter values, and each sample is used to run the CAM model to calculate total precipitation.

In this work, 60 samples are extracted from the 6 parameters described in Section 2.2 using the Latin hypercube sampling method, and the steps are as follows. For one-dimensional Latin hypercube sampling, the cumulative density function is divided into $n$ equal partitions, and a random data point is chosen within each partition. We know that each parameter is uniformly

distributed within its value range. To obtain 60 samples, each parameter is divided into 60 groups without overlapping in the definition range based on the cumulative density function of uniform distribution. The probability of each group being obtained is 1/60. In the interval of each group of parameters, one parameter value is selected randomly. There are 6 vectors generated by the above rules, each of which represents the sampling results of one parameter and contains 60 elements. We try to obtain a 6*60 matrix, where each row of the matrix represents a sample point, so that we randomly select one element in each vector

and form a new 6-dimensional vector. There are 60 vectors in total. The sampling points are evenly distributed throughout the solution space.

### 3.2.2  Fitness function

The objective of parameter tuning is to enhance the CAM simulations, leading to a better alignment with the reanalysis data. Therefore, the root-mean-square error (RMSE) is used in this study to evaluate discrepancies between the model simulation

and observation data. The RMSE is computed by comparing the meteorological variable total precipitation (PRECT) with the observation data, and the expression of RMSE is as follows:

$$RMSE = \sqrt{\frac{1}{N}\sum_{i=1}^{N}(mod_i - obs_i)^2} \qquad (1)$$





where $N$ is the total number of grid points in the simulation region and $mod_i$ and $obs_i$ are the model-simulated and observation data values at grid point $i$, respectively. A smaller RMSE value means a smaller error between the model simulation and observation data. The objective of the optimization method is to minimize the RMSE of each region.

### 3.2.3 Generate new sample points

After completing the above steps, a surrogate model for the CAM precipitation simulation results is constructed. Generally, when solving a complex parameter optimization problem by a surrogate model, to improve the accuracy simulation results of the surrogate model, additional new sample points need to be added. new sample points need to be added to improve the accuracy of the surrogate model, thus reducing the number of simulations of the actual complex model. Strategies for generating new sample points transform the process of parameter point generation into optimization problems, employing an evaluation criterion. They are iterative methods and the new parameter point generations are guided by utilizing data acquired from previous iterations. In this step, a new parameter set will be created for the optimal parameter values of the global-level surrogate model, and the optimal parameter values will be put into the CAM to obtain the corresponding RMSE between this simulation and the observation data.

There are several methods to generate the next point sample point, for example, minimum interpolating surface (MIS) strategy (Jones, 2001) and maximizing expected improvement (MEI) strategy (Jones et al., 1998). In MIS strategy, after identifying the minimum of the surrogate model, it is considered as a new point and added to the initial sample set, and then the surrogate model is updated. However, the MIS strategy is suitable for surrogates with high accuracy. CAM is a complex ESM with strong nonlinearity, and the surrogate model constructed by dozens of sampling points cannot maintain a high accuracy. The MEI strategy utilizes the standard error to encourage the method to revisit the regions with sparse points. This method requires computing the expected improvement resulting from sampling at a specific point.. However, it is easy to fall into a local minimum.

Therefore, we need an appropriate strategy to update the surrogate model for CAM precipitation. In this work, we select the candidate point (CAND) approach (Regis and Shoemaker, 2007), which achieves a balance between exploitation and exploration. Exploitation refers to the rapid convergence within a region to identify a local optimum. Exploration involves searching in uncharted regions of the parameter space to locate a global optimum. Balancing the exploration and exploitation ensures that the tuning method can find the genuine global optimum and avoids wasting computational cost on meaningless parameter areas. The CAND approach is as follows.

There are four steps in the CAND strategy. First, the maximum and minimum of the surrogate model are calculated. A candidate point set $\Omega$ is generated, and the corresponding fitness function values of each point are calculated and represented by $s(x)$.

In the second step, the $V^S$ values are calculated, and the $V^S$ values represent the exploitation mechanism, which describes the level of the fitness function values of each point in the candidate point set $\Omega$ output by the surrogate model. A smaller $V^S$ value means a better simulation result of the current surrogate model.





In the third step, the $V^D$ values of each point in $\Omega$ are calculated. A smaller value of $V^D$ means that the current point is an effective exploration. The $V^D$ values represent the exploration mechanism. A smaller value of $V^D$ means that the current point is an effective exploration. $\delta(x)$ represents the minimum distance from point $x$ in $\Omega$ to the point in the initial sample set generated in Section 3.2. The $V^D$ values express the exploration for the unknown region of the current surrogate model. We know that the sample set used to construct the surrogate model is just a small part of the entire parameter space. Using the surrogate model to quickly traverse the unknown region is conducive to obtaining higher quality solutions in these unknown regions.

In the fourth step, for each point $x$ in $\Omega$, the values of the acquisition function $\omega V^S(x) + (1 - \omega) V^D(x)$ are calculated, and the minimum acquisition function value is selected from the candidate point set $\Omega$. $\omega$ represents an accommodation coefficient that balances the local optimum and global optimum. Corresponding point coordinates will be sent into the actual complex model in this CAM5.3 study to obtain the RMSE of this set of parameters. The candidate point $x$, and its RMSE, will be sent to the initial sample set generated in Section 3.2, the surrogate will update based on the new sample set, and this iteration is then finished. After each iteration, a new pair of parameters their corresponding RMSE will be incorporated into the initial sample set. After an initial sample set updating the surrogate model will update.

In the CAND approach, the MIS strategy is used for exploitation. The strategy for exploration is calculating a weight parameter which is represented by a distance between the candidate point and the sample points. The set of the previously sampled points denote the region which has been explored, allowing us to estimate uncertainty from the weight parameter based on the distance to this region. To generate the new sample point, another weighted combination of these two strategies is employed. Our operations and calculations in the CAND strategy are based on the surrogate model, which greatly reduces the number of CAM executions in the parameter tuning process. We can take advantage of the surrogate model; even if the candidate sample set $\Omega$ is very large, the surrogate model can still obtain the RMSE results in a very short time.

### 3.3 Local-level surrogate model

### 3.3.1 Surrogate model construction

Because the surrogate model is constructed by a small number of real complex model samples, it cannot accurately simulate the actual situation of the complex model. Moreover, the optimal solution determined by the surrogate model is an approximate value, and the approximate performance of the surrogate model within the local range near the optimal solution is closely related to the overall optimization accuracy. Therefore, it is necessary to establish a local-level surrogate model to mine the local optimal value of the real complex model so that the local approximation ability is enhanced.

The Gaussian process (GP) is used to construct the local-level surrogate model in this study. GP can transform discrete point distributions into function distributions, and is more adapted to small-scale optimizations (Karim et al., 2020). GP is defined as a prior distribution over function, and the original model $f(x)$ is assumed generated from such a prior distribution (Garrido-Merchán and Hernández-Lobato, 2020). A GP is characterized by the mean function $m(x)$ and its covariance function





$k(x,x')$:

$$m(x) = E[f(x)] \tag{2}$$

$$k(x,x') = E[(f(x) - m(x))(f(x') - m(x'))] \tag{3}$$

For the regression problem, consider the following model:

$$y = f(x) + \epsilon \tag{4}$$

where $x$ is the input vector and $f$ is the output variable. $y$ is the training output corrupted by the additive noise, and $\epsilon$ represents the independent and identically distributed Gaussian noise term $\epsilon \sim N(0, \sigma_n^2)$ with 0 mean and $\sigma_n^2$ variance. The prior distribution of $y$ is as follows:

$$y \sim N(0, K(X,X) + \sigma_n^2 I_n) \tag{5}$$

We can obtain the joint distribution of training outputs $y$ and the testing outputs as a joint Gaussian distribution:

$$\begin{bmatrix} y \\ f_* \end{bmatrix} \sim N \left( 0, \begin{bmatrix} K(X,X) + \sigma_n^2 I_n & K(X,X_*) \\ K(X_*,X) & K(X_*,X_*) \end{bmatrix} \right) \tag{6}$$

where $f_*$ and $X_*$ respectively represents the test output and test input matrix, and $K(X,X)$ is the covariance matrices of the training inputs. $K(X,X_*)$ is the covariance matrices between the training inputs and test inputs and $K(X_*,X_*)$ represents covariance matrices of the test inputs. Thus, we can deduct the posterior probability distribution of these test outputs $f_*$:

$$f_* \mid X,y,X_* \sim N(\overline{f_*}, cov(f_*)) \tag{7}$$

$$\overline{f_*} = K(X_*,X)[K(X,X) + \sigma_n^2 I_n]^{-1} y \tag{8}$$

$$cov(f_*) = K(X,X_*) - K(X,X_*)[K(X,X) + \sigma_n^2 I_n]^{-1} K(X,X_*) \tag{9}$$

The mean and variance of the predicted values for the test point $X_*$ are denoted as $\widehat{\mu_*} = \overline{f_*}$ and $\widehat{\sigma_{f_*}^2} = cov(f_*)$, respectively.

### 3.3.2  Trust Region Method

The trust region method is utilized to solve the unconstrained optimization problems. For a continuous differentiable function of the second order, the primary concept revolves around approximating the objective function with a quadratic function in the vicinity of the extreme point. This quadratic function seeks optimality within the current trust region centered around the optimal point. The range of the trust region will be dynamically adjusted based on the ratio of the real function to the quadratic function.

In this paper, CAM parameter tuning is a nonlinear optimization problem that is difficult to describe by a mathematical function; therefore, the surrogate model is built as a replacement for the quadratic function. and the ratio is calculated as follows:

$$\sigma = \frac{f(x_{t+1}) - f(x_t)}{f(\hat{x_{t+1}}) - f(\hat{x_t})} \tag{10}$$





where $f(x_t)$ is the real model output, in this problem it is the CAM simulation result corresponding to the solution which is the near optimal in the t-th iteration., and $f(\hat{x}_t)$ is the estimated value of the near optimal solution obtained by the surrogate model. The ratio is represented as follows:

$$\Delta_{k+1} = \begin{cases} 0.25\Delta_k & \sigma \le 0.25 \\ \Delta_k & 0.25 < \sigma \le 0.75 \\ 2\Delta_k & \sigma > 0.75 \end{cases} \quad (11)$$

The range of the new trust region depends on the quality of the surrogate model fitting result. If the fitting result is good, the trust region will be expanded or kept unchanged in the next iteration; otherwise, the trust region will be reduced. When $\sigma$ is less than 0.25, the fitting result is not accurate within the current range, and the trust region should be reduced. When $\sigma$ is close to 1, which represents the high proximity of the surrogate model, we can expand the radius of the current trust region; otherwise, the trust region keeps the original range in this iteration.

The trust region-based process in this proposed method is shown in Algorithm 2:

---

**Algorithm 2** The trust region method

---

1: Select initial point $x_0$ and intial range of trust region.

2: **while** End condition is not met **do**

3:     Calculate the $\sigma$ value, update the trust range.

4:     Add the $x_t$ and $f(x_t)$ to the points set of the local-level surrogate model and update the model.

5: **end while**

---

In this method, we select the best ten percent sampling points from the sample set of the global-level surrogate model to construct the Gauss process model. The optimal solution value generated by the global-level surrogate model is set as the initial point, and the maximum space enclosed by the initial point set of the local-level surrogate model is defined as the initial trust region.

## 4 Result

### 4.1 Method evaluation

#### 4.1.1 Surrogate Model Construction

Different methods are selected to construct surrogate models for different types of problems. We evaluate which of the following surrogate model construction methods are the most suitable for a CAM precipitation simulation: (a) Adaboost, (b) random forest (RF), (c) support vector machine (SVM), (d) bagging, (e) gradient boosting regression tree (GBRT), (f) decision tree and (g) K-nearest neighbor (KNN). Cross validation (CV) (Browne, 2000) is a statistical method used to compare each machine learning algorithm. In this paper, the N-fold CV method is implemented to obtain the best surrogate model method for simulating CAM precipitation, which works as follows. The entire sample set $S\{X, Y\}$ is divided into $N$ subsets. Each of them is

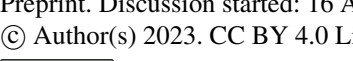



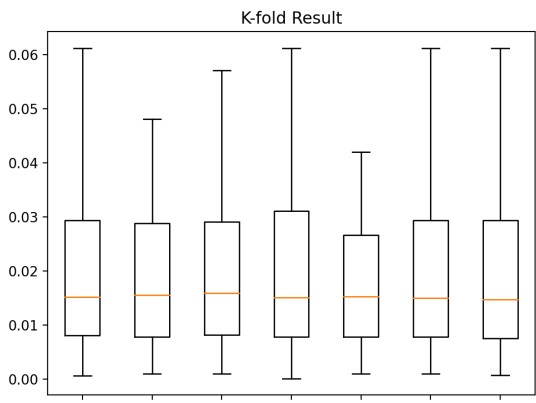

**Figure 2.** Cross-validation results to compare the performance between these methods. The Y-axis represents the relative error between the experimental and reanalysis data.

equal and independent.

$$S\{X,Y\} = S^1\{X,Y\}, S^2\{X,Y\}, ..., S^n\{X,Y\}. \tag{12}$$

With a specific surrogate model method, the surrogate model is constructed by a $N-1$ subset of points. The input data are these subsets, and the output data of the surrogate model are the RMSE results of the corresponding sample points. The surrogate model is built $N$ times. For each time, $N-1$ subsets are used to create the surrogate model for training datasets. The remaining dataset is regarded as the test dataset. The relative error is used to evaluate the performance of these methods in this paper. The relative error is calculated between the simulation results obtained by the surrogate model built on the $N-1$ subsets and the results in the test data. After $N$ iterations, the relative error can be obtained as the evaluation criterion to compare the performance of each method of surrogate model construction. The cross-validation result is shown in Figure 2. Compared with other methods, the overall error of the GBRT-based model is lower, and the error distribution space is more concentrated. Therefore, the GBRT method is selected to construct the global-level surrogate model.

### 4.1.2 Comparison of the optimization processes

To prove the tuning effect of the proposed method, we design an experiment to compare this method with an ASMO method based on the GP and MIS strategy. The optimization process of the two methods is shown in Figure 3. Compared with the ASMO optimization method, we improve the optimization efficiency in many ways. Both the convergence speed and simulation accuracy of the surrogate model are improved by the proposed multilevel surrogate model-based method. The CAND strategy can effectively balance exploration and exploitation when updating the surrogate model so that the model can receive the trend and direction of the better solution. The multilevel surrogate model fully considers the existence and quality of these

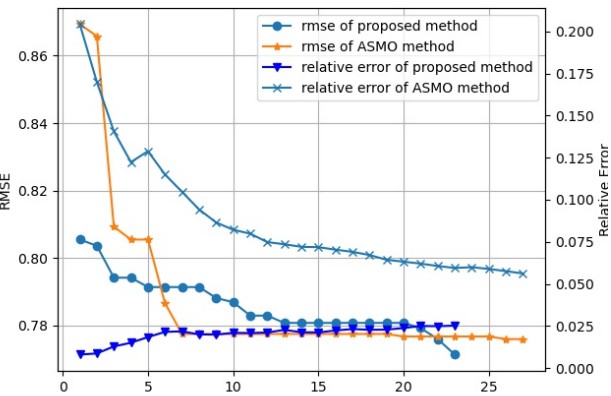

**Figure 3.** The tuning process with the proposed multilevel surrogate model-based method and ASMO method, and the corresponding relative error of these two methods, where the relative error is equal to $(|predictvalue - realvalue|)/realvalue$.

local optima, and explores the regions where the global surrogate model is difficult to simulate or the simulation effect is not accurate enough to obtain new optimal solutions.

### 4.1.3 Efficiency analysis

As mentioned earlier, one of the biggest advantages of the surrogate model is to reduce the computational cost. In this method, this advantage is represented in two aspects. For each iteration of each optimization step, we only run the real model once, and all of the samples are predicted using the surrogate model. The reduction in the computational cost is directly proportional to the number of samples generated in each step. In the entire optimization process, the number of iterations is less than that of the other methods (Figure 3). Unlike some intelligent algorithms, this method does not have the concept of "individuals", so

there will not be a large amount of redundant computation during the optimization process.

### 4.2 The limitations of global optimization

In the optimization experiment of constructing the surrogate model (Figure 4), we found that the global tuning based on the surrogate model does not change much compared with the default value, which shows that the surrogate model has some limitations in the process of CAM global optimization. The complexity is related to the nonlinear interactions of the earth

system model. However, the global optimum does not mean that it is the optimal result in each region. there is still an optimization space. Therefore, we need to further research this limitation and determine the most suitable parameter optimization use scenario for the surrogate model.

     To give full play to the best performance of the surrogate model, we research the relationship between the parameter change and result change in CAM precipitation, explore the influence of a single parameter disturbance on the result and compare it

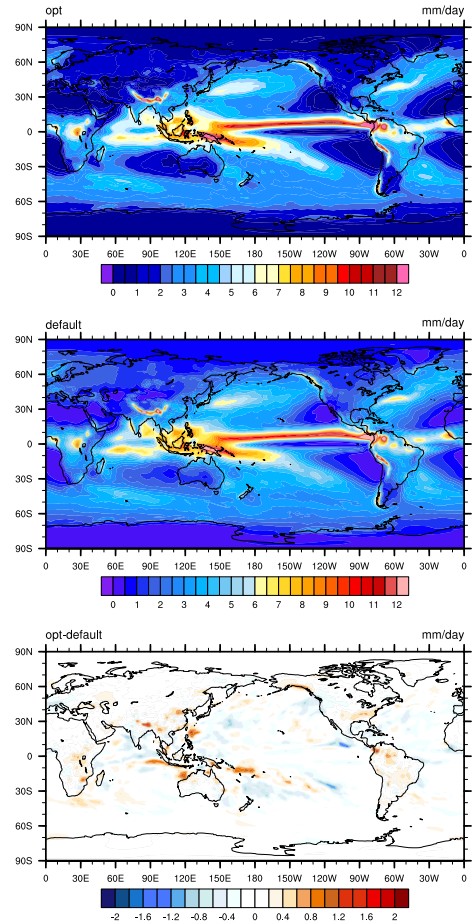

**Figure 4.** The precipitation distribution of global optimization, there are optimal experiment, default simulation difference between optimal experiment and default simulation from top to bottom.

with the default precipitation value; thus, the influence mode and intensity of parameter changes on global precipitation are obtained. Using the samples generated in Step 1 of Section 3, we first calculate the difference between the disturbance value and the default value of 60 groups of single parameter samples. Then, a short-term hindcast named the cloud-associated parameterizations Trestbed (CAPT) (Xie et al., 2004) with an interval day 3 hindcast proposed by (Zhang et al., 2018) is simulated 60 times to obtain the precipitation value corresponding to the disturbance parameters, and to calculate the difference with

the default precipitation value. Pearson correlation analysis is carried out between them to calculate the symbolic consistency of the two groups of differences to determine the impact of the same parameter value on different regions. Taking rhminl as an example, the results are shown in Figure 5. The impact of parameter changes on the world is different. There is a strong negative correlation in marine regions, such as the South and North Pacific, the Indian Ocean and the Atlantic, while there is a positive correlation on land, mainly concentrated in Eurasia and South America, Central Africa and other regions. This



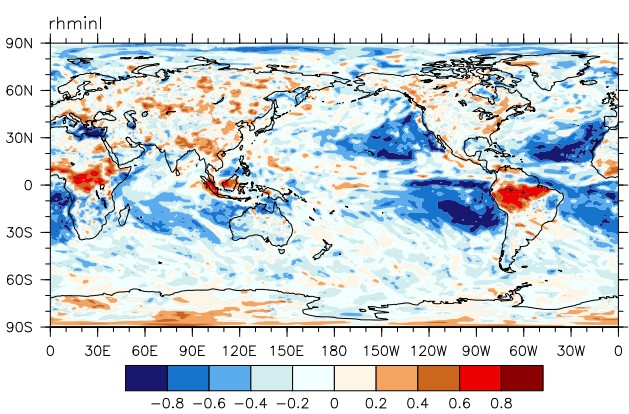

**Figure 5.** The symbolic consistency of rhminl disturbance and corresponding precipitation change.

**Table 3.** The CAM simulation performance increase for each region.

| Region | Default RMSE | Optimized RMSE | Reduction Rate |
|---|---|---|---|
| WarmPool | 1.985 | 1.924 | 3.07% |
| South Pacific | 0.855 | 0.455 | 46.78% |
| Niño | 0.931 | 0.773 | 17.04% |
| South America | 2.576 | 2.371 | 7.94% |
| South Asia | 1.484 | 1.293 | 12.87% |
| East Asia | 1.213 | 0.878 | 27.68% |

means that when the rhminl value is disturbed relative to the default value, the change in precipitation in some regions may be completely opposite, similar to a "rocker" effect. In such a situation, it is difficult to use one parameter value to optimize global change, and the increase in parameter numbers will further enhance this rocker effect, making the mapping between parameters and precipitation complicated. To some extent, the surrogate model has difficulty approximating this complex change through a limited number of samples and iterations, which will greatly reduce the surrogate model optimization performance.

The influence of the parameters on different regions, and the simulation results of each region are considered. We choose region-based optimization, abide by the parameter disturbance characteristics of the model, and give full play to the performance advantages of the surrogate model as much as possible.





## 4.3 Region-based optimization result

Based on Section 4.2, because of the "rocker" effect in CAM, the optimization result of the region may be inconsistent with
the global optimization result. Therefore, region-based optimization is more suitable than global optimization. Considering
the global precipitation distribution, six regions are selected in this study in Table 2; they are WarmPool, South Pacific, Niño,
South America, South Asia and East Asia. A surrogate model is constructed for each region based on Section 3. The RMSE
of each region is calculated according to the range shown in Table 2 in the corresponding optimization process. Altogether,
6 surrogate models are constructed for the selected regions, and each region is optimized separately. The region optimization
395 result is shown in Table 3. The simulation result in each region has been advanced to different degrees, and the results of the
South Pacific, East Asia, Niño and South Asia are notably improved. We will discuss these results.

The observation data show that the precipitation in the South Pacific region generally reveals a ladder-like decline from east to
west. The default experiment can simulate the precipitation change trend; however, there is a huge deviation in the precipitation
value during the precipitation decline in the default simulation. In Figure 6, in the observation data, the precipitation in most
400 areas is less than 3 mm/day. In the default experiment, the precipitation in many areas is over 3 mm/day, which leads to the
fact that the precipitation values obtained by the default simulation are larger than the observation data, and the xy plot of the
zonal mean in Figure 7 can also capture the large difference between them. The optimization experiment maintains the change
trend of the default experiment, which is consistent with the observed data, and reduces the area in which precipitation is over
3 mm/day. This positive change is more obvious in the areas far from the equator. In the southeast, the error is reduced by 90%,
from 2 mm/day to approximately 0.2 mm/day. Moreover, the optimization did not cause a new deviation. The xy plot of the
zonal mean shows that the optimization results over 26°-4°S are significantly better than the default experiment. Although it
did not bring a significant improvement over the two edges, the precipitation remained at a level, almost equal to the default
value, without causing a new error. From a macro perspective, the improvement of RMSE also reaches 46.78%, which is the
most significant effect in the region selected in this study.

Compared with several other optimization areas, the WarmPool area spans multiple longitude and latitude ranges, the oceans
and continents are intertwined, and the precipitation is relatively large but the optimization results are still positive in many
areas. In the south latitude area, there is a phenomenon of heavy precipitation inconsistent with the observed data in the default
experiment, and the optimized results in Figure 8 show that the precipitation in these areas has been effectively weakened in the
model simulation, and the result is closer to the observation data. This improvement can be clearly shown in the xy plot of the
zonal mean in Figure 9. Furthermore, the problem of less precipitation in the default simulation over Sulawesi Island has been
solved, and the regions centered on the island have been positively changed. The improvement in the Northern Hemisphere
is mainly concentrated in the areas of 4°-10°N and 130°-150°E. The precipitation intensity of the default simulation in this
area is far less than the observation data, and the optimization experiment increases the precipitation in this area. The situation
of excessive precipitation in the northwest of the WarmPool has also been improved to a certain extent in the optimization
experiment. However, in areas greater than 10°N, the simulation results of the optimization experiment are not ideal, which
also makes the deviation of the corresponding areas in the XY plot of the zonal mean larger and leads to an insufficient

—

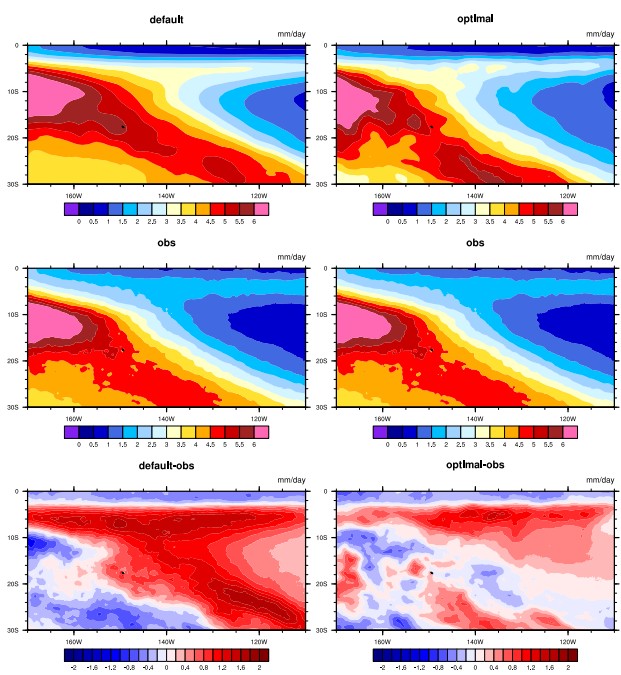

**Figure 6.** The precipitation distribution of the South Pacific optimization result. The left column shows the default simulation, observation data and difference between the default simulation and observation data from top to bottom. The right column shows the optimal experiment, observation data and the difference between the optimal experiment and the observation data from top to bottom.

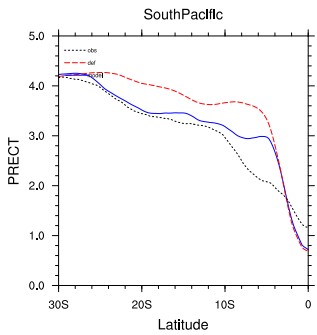

**Figure 7.** The xy plot of the zonal mean of precipitation over the South Pacific; the X-axis shows the latitude, and the Y-axis represents the precipitation value. The blue, red, black lines represent the optimal experiment, the default simulation and the observation data, respectively.

improvement in the RMSE. Nevertheless, for the WarmPool, the optimization results still reflect a change trend consistent with the observation.

The XY plot of the zonal mean of South Asia is shown in Figure 10, and it can be seen that the tendency of precipitation with the latitude has changed completely after optimization compared with the default simulation. The overall precipitation of



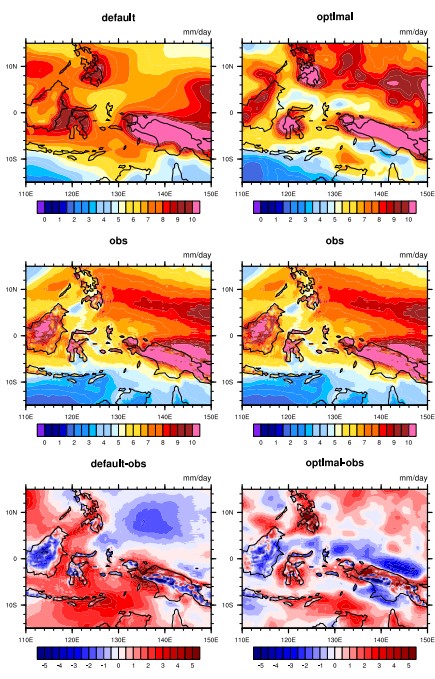

**Figure 8.** The precipitation distribution of the WarmPool optimization result. The left column shows the default simulation, observation data and the difference between the default simulation and the observation data from top to bottom. The right column shows the optimal experiment, observation data and the difference between the optimal experiment and the observation data from top to bottom.

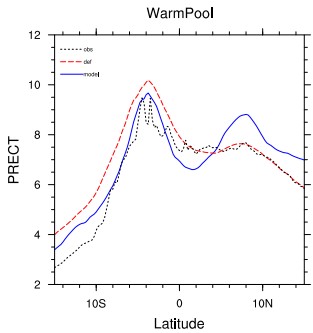

**Figure 9.** The xy plot of the zonal mean of precipitation over WarmPool; the X-axis shows the latitude, and the Y-axis represents the precipitation value. The blue, red and black lines represent the optimal experiment, the default simulation and the observation data, respectively.

the default experiment presents an upward trend with the latitude and expresses a downward trend after optimization, which coincides with the reanalysis data, and the optimized value is closer to the reanalysis data than the default simulation. The precipitation distribution of the default simulation and optimization result are shown in Figure 11. Compared with the default simulation, ocean precipitation is increased over western Indonesia so that the negative error is almost eliminated, and it is





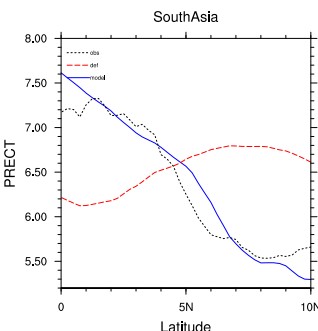

**Figure 10.** The xy plot of the zonal mean of precipitation over South Asia; the X-axis shows the latitude, and the Y-axis represents the precipitation value. The blue, red and black lines represent the optimal experiment, the default simulation and the observation data, respectively.

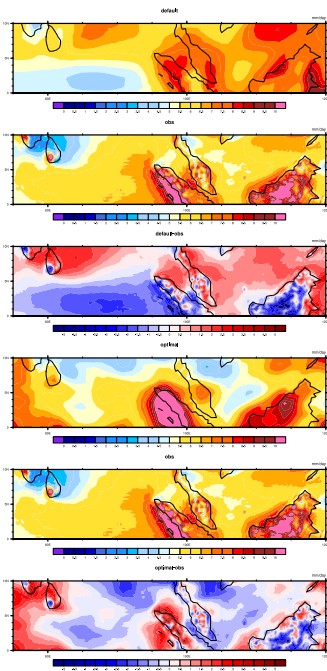

**Figure 11.** The precipitation distribution of the South Asia optimization result includes a default simulation, observation data, difference between default simulation and observation data, optimal experiment, observation data and the difference between the optimal experiment and the observation data from top to bottom.

decreased over 105-120°E. For land precipitation, the changes are smaller than for ocean precipitation, and there also exist differences in magnitude. However, the error between simulation and observation is significantly reduced.

    In the default simulation of Niño region (Figure 12), the observation inconsistency is mainly concentrated in two areas, the negative error area on the left and the positive error area on the right. For the improvement and optimization results of



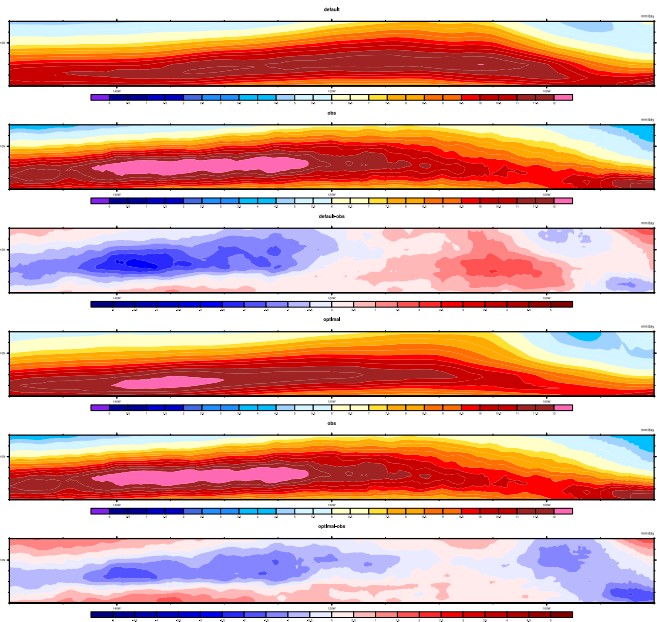

**Figure 12.** The precipitation distribution of the Niño optimization result includes default simulation, observation data, difference between default simulation and observation data, optimal experiment, observation data, and difference between optimal experiment and observation data from top to bottom.

these two areas, there are obvious heavy rainfall centers; the positive error area is basically consistent with the observation data, and the negative error area is also reduced. The central region with a large error generated by the default experiment still exists after optimization but the value is closer to the observation data than the default result. On the whole, it shows a positive optimization, however, in the XY plot of the zonal mean in Figure 13, the precipitation over the high value area in the optimization result is lower than the observation. The reason is, that when improving the area of positive difference, a new small part of the negative error is introduced from the area east of 100°W, which is the reason for the large difference in the zonal mean between the simulation and default.

The results in East Asia are similar to those in South Asia, which is shown in Figure 14. For the precipitation in East Asia, the default experiment is generally less than the reanalysis data, and the obvious improvement is in southern China and the Huanghai Sea region. We can see that in the default experiment, there is a relatively large error over the Donghai Sea, centered around the area between Taiwan and Japan, even extending to the Huanghai Sea, the Sea of Japan and the Korean Peninsula. The optimization results suggest that this error is almost eliminated over these regions, and it is also markedly reduced over the central area. The same is true in southern China. Precipitation in southern China is increased after parameter optimization, which is low in default simulation and hard to match with observations. In addition, other areas are also improved to different degrees, such as central China and the South China Sea. We observe such positive changes using the XY plot of the zonal mean





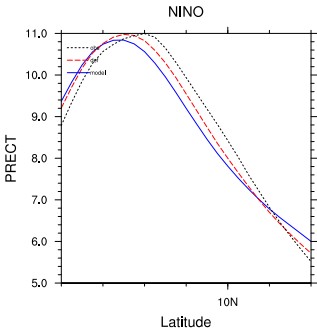

**Figure 13.** The xy plot of the zonal mean of the precipitation over Niño, the X-axis shows the latitude, and the Y-axis represents the precipitation value. The blue, red and black lines represent the optimal experiment, the default simulation and the observation data, respectively.

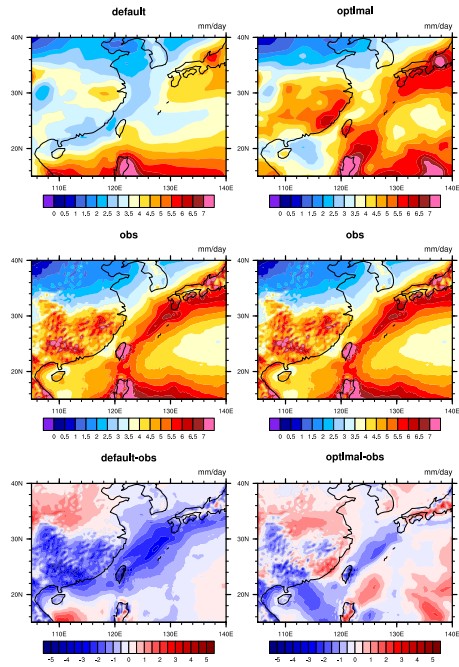

**Figure 14.** The precipitation distribution of the East Asia optimization result. The left column shows the default simulation, observation data and the difference between the default simulation and the observation data from top to bottom. The right column shows the optimal experiment, observation data and the difference between the optimal experiment and the observation data from top to bottom.

(Figure 15), which demonstrates that the optimization result is better than the default simulation in most cases, especially at 450    20-30°N, which corresponds to the changes over southern China and the Donghai Sea.

As shown in Figure 16, the characteristics of the precipitation in South America are wide ranging and high in quantity, and it is difficult to obtain results consistent with the observations. Compared with the default, the optimized results reflect



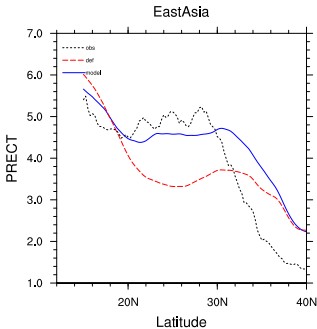

**Figure 15.** The xy plot of the zonal mean of precipitation over East Asia; the X-axis shows the latitude, and the Y-axis represents the precipitation value. The blue, red and black lines represent the optimal experiment, the default simulation and the observation data, respectively.

precipitation changes that are more consistent with the observations in some areas. The zonal mean (Figure 17) shows that the optimization result has a better performance at 12°S-2°N, and the spatial distribution shows that a small area at approximately 65°-60°W near the south of the equator has been improved after a parameter optimization. Even so, there is a great difference between the model simulation and the observation when evaluating South America. Overall, the optimization effect is not significant.

### 4.4 Ensemble optimization results of the nonuniform parameter parameterization scheme

In the local optimization experiments, we found that a local optimization may lead to the simulation results of other regions moving in a worse direction, that is, the optimization of one region is at the cost of worse results in other regions, which is unacceptable. Therefore, we design a nonuniform parameter parameterization scheme to integrate the optimal parameters of multiple regions into one case. In the selected region, we use the parameters obtained through the surrogate model-based optimization methods, and use default values in other regions. To prevent the large difference in parameter values between the two sides of the regional boundary from affecting the experimental results, we design a boundary smoothing scheme to achieve a smooth transition of parameter values from the center to the boundary of each region. The same or similar parameter values are obtained at the boundary:

$$x = max(\left|\frac{lat - centerlat}{\frac{1}{2}width}\right|, \left|\frac{lon - centerlon}{\frac{1}{2}length}\right|) \tag{13}$$

$$weight = \frac{1}{2}(cos(\pi x^3) + 1) \tag{14}$$

$$value = (optmization\_value - default\_value) * weight + default\_value \tag{15}$$

where $x$ is used to evaluate the distance from each point in the region to the center of the region. $lat, lon$ represent the latitude and longitude of the point, respectively, and $centerlat, centerlon$ represent the latitude and longitude of the center point. $width, length$ represent the width and length of the selected region. The denominator is equal to the distance between two boundaries and the center point. $x$ means the maximum distance from the point to the length or width of the region, and the





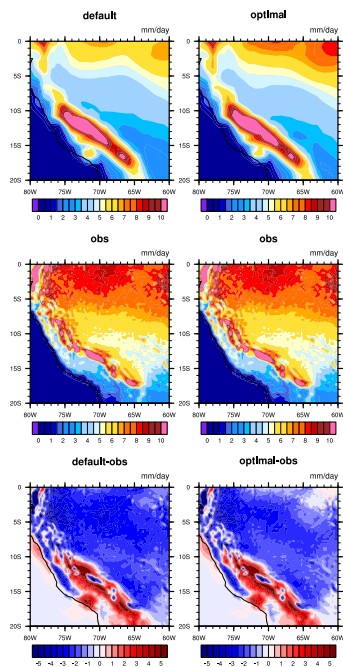

**Figure 16.** The precipitation distribution of the South America optimization result. The left column shows the default simulation, observation data and the difference between the default simulation and the observation data from top to bottom. The right column shows the optimal experiment, observation data and the difference between the optimal experiment and the observation data from top to bottom.

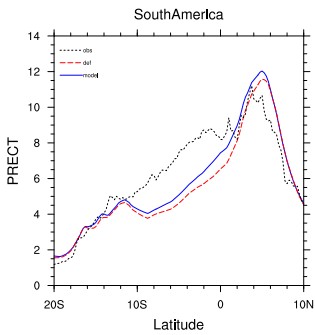

**Figure 17.** The xy plot of the zonal mean of precipitation over South America; the X-axis shows the latitude, and the Y-axis represents the precipitation value. The blue, red and black lines represent the optimal experiment, the default simulation and the observation data, respectively.

value is between 0 and 1. $weight$ is the weight of the distance for each point. The closer to the center point, the closer the
weight is to 1, and the closer the parameter value is to the optimized value. The closer to the boundary, the closer the weight is to 0, and the closer the parameter value is to the default value. Equation 15 provides a cosine weight function, which allows



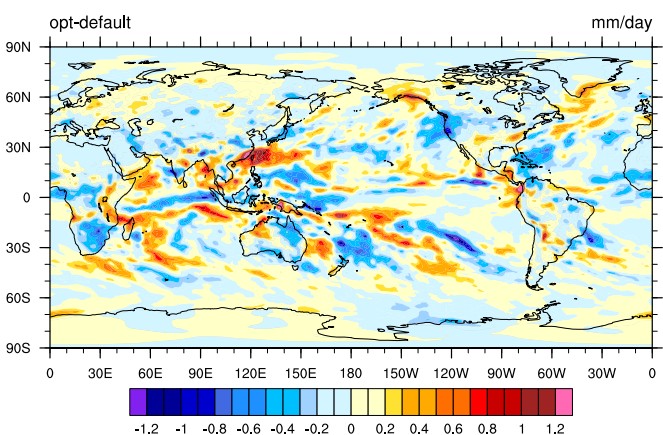

**Figure 18.** The difference between the experiment simulation result with the ensemble parameters of each region and the default experiment simulation result

each real number between 0 and 1 to correspond to a 0-1 variation interval, and the dependent variable gradually decreases as the independent variable increases.

The ensemble parameter experiment results are shown in Table 4. In this case, different parameters were set according to
different regions, and out of the six regions, four regions achieved better simulation results than the default experiment, with two regions performing slightly worse than the default experiment. We know that the simulation results of each region are to some extent influenced by the parameter values of other regions, so it is difficult to achieve a level of individual optimization for each region in Section 4.2. Among them, the South Pacific, Niño, South America and East Asia regions all achieved varying degrees of improvement, while the WarmPool and South Asia regions slightly decreased compared to the default values. The
distribution map of precipitation is shown in Figure 18. Compared with the default experiment, the new experimental results have undergone many positive changes; it reduced precipitation in North China and increased precipitation in the East China Sea and the ocean area east of the East China Sea, which are the reasons for the improvement of the simulation results over East Asia. In the Pacific region, a portion of the precipitation within the 150W-120W region was reduced, however, there was an increase in precipitation in the western region and near the equator, which also resulted in the simulation results in the South
Pacific region not being as significant as the regional optimization results in Section 4.2. The precipitation levels in the Niño region located in the eastern Pacific Ocean near Central America are reduced, and the improvements in some areas to the west are all positive. On the South American continent, some small-scale increases in precipitation are the reason for improving the



**Table 4.** The tuning results of the ensemble parameter experiment over a selected region.

| Region | Default RMSE | Optimized RMSE | Better(+) or Worse(-) |
|---|---|---|---|
| WarmPool | 1.985 | 2.037 | - |
| South Pacific | 0.855 | 0.802 | + |
| Niño | 0.931 | 0.741 | + |
| South America | 2.576 | 2.540 | + |
| South Asia | 1.521 | 1.759 | - |
| East Asia | 1.213 | 1.054 | + |

simulation effect in South America, as such areas are relatively small, so the overall improvement is not significant. In addition, in regions such as Southern Africa, Central Asia and the Gulf of Mexico, precipitation decreases to varying degrees compared

to the default experiments. However, the increase in precipitation in the WarmPool and South Asian regions result in a slight increase in error between the simulation results and the default experiment. Overall, the results of such simulation experiments can reach a level close to the default experiment in most regions, with relatively positive changes in some regions, and only a small portion of the results are slightly lower than the level of the default experiment.

## 5 Conclusions

The surrogate model is widely used to solve complex and expensive optimization problems, such as various earth system models, and it has not received enough attention in CAM parameter optimization. In this paper, we propose a multilevel surrogate model-based parameter optimization method for CAM precipitation. First, we demonstrate that the surrogate model method can be used to improve the precipitation simulation of CAM. An appropriate surrogate model is selected by cross validation. Second, we propose a multilevel surrogate method, which is demonstrated to have a faster convergence speed

and fewer errors during the tuning process. Third, because of the "rocker" effect of the parameter, we design region-based optimization experiments and select six regions as the performance indicators. The improvement is more pronounced in the South Pacific and East Asia regions, reaching 46.78% and 27.68%, respectively. The improvement in the Niño and South Asia regions also reach more than 15%. The two regions with a more modest improvement are the WarmPool and South America regions, which are both below 10% at 7.41% and 7.94%, respectively. To integrate these results over each region, we attempt

a nonuniform parameter parameterization scheme and implement weight-based boundary smoothing for each region. In the new parameterization scheme, the same parameter has different values in different regions. The optimal parameters of the six regions are integrated into one case over the corresponding region. The ensemble parameter experimental results showed positive changes in four of the six regions we selected but two regions did not improve. This also reflects certain limitations in the CAM surrogate model-based optimization. In future work, the scope of the application and construction methods of



the surrogate models will be further explored to improve the efficiency of CAM optimizations. In addition, we will apply this
method to other variables or for multiobjective optimizations.

*Code and data availability.*   The code of our algorithm, the observations, and the related scripts can be found at
Xianwei, Wu. (2023). Surrogate model-based precipitation. Zenodo. https://doi.org/10.5281/zenodo.8205258

*Author contributions.*   XW and JZ proposed the tuning method. LH and LW evaluated the experiment results. HL provided assistance and
support with the experiments. The paper was written by XW, JZ, LH, LW, HL.

*Competing interests.*   The authors declare that they have no conflict of interest.

*Acknowledgements.*   This work is funded by: National Key RD Plan of China under Grant No. 2017YFA0604500, Key scientific and techno-
logical R&D Plan of Jilin Province of China under Grant No. 20180201103GX, the National Key Research and Development Plan of China
under Grant No. 2020YFB0204800, the National Natural Science Foundation of China under Grant No. T2125006, the Jiangsu Innovation
Capacity Building Program under Grant No. BM2022028.



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
