# Peer review of "Surrogate model-based precipitation tuning for CAM5"

_Geoscientific Model Development, 2023_

## Author Comment (AC1)

**Replies to Referee #1, GMD-2023-164**

Xianwei Wu, Liang Hu, Lanning Wang, Haitian Lu, and Juepeng Zheng

November 15, 2023

Thank you very much for your patient and detailed comments on our work [1]. These valuable comments are very helpful for us to improve this paper. After carefully reading all the questions, we have answered each of them and will make appropriate corrections in the revised version of our manuscript.

In this attachment, the blue paragraphs represent your comments, and the black paragraphs below are our corresponding replies.

**1 Replies to major comments**

Replies to major comments are as follows:

1. The manuscript structure, particularly the method section, needs to be reorganized to improve the compactness. There are several areas that require clarification. For instance, Algorithm 1 calculates the RMSE, but its definition is found in section 3.2.2. It would be more appropriate to move the definition to section 2. Additionally, in Line 4 of Algorithm 2, it is unclear whether the new parameters are obtained using CAND. Furthermore, it is not explained why the local-level surrogate utilizes Gaussian Process. In addition, it could describes the difference between the algorithm used in this work and the ASMO. Typically, optimization algorithms require hundreds of steps to achieve convergence, but in this work, only around 20 steps of local optimization are performed. It is hard to say the algorithms get convergence. It appears that the ASMO method can achieve local optimization more quickly. The conclusion is not convinced. The description of CAND is difficult to follow, particularly the calculation vs and vd, which is lack of calculation details. The cross validation describe can move from result section to the method section.

This comment contains multiple questions, we will reply these questions separately.

1).We will improve the structure of the study in revised manuscript according to the comments.

2).The new parameters are not obtained by CAND, CAND is just used for global-level surrogate model.

3).Unlike global-level surrogate-model, the local-level surrogate model is required to have a higher accuracy with fewer samples. So that construction method of the loacl model is more important. according to [2, 3] GP is selected to construct local-level surrogate.

Similar to the global-level surrogate model, we conduct a cross-validation for several method to construct local surrogate model, the results are shown in Figure 1:

We will add these results to the rivised manuscript.

4). The entire optimization process consists of over 20 steps. After obtaining the current optimal solution, there are several validation steps. Once these validation steps are completed, and no new optimal solution is found, we consider the current optimal solution as the final result of the optimization. In the figure, we illustrate the reduction in RMSE during the optimization process and the associated errors.

In terms of errors, it's possible that our method may indeed have slightly higher errors compared to ASMO in the end. However, our method demonstrates greater stability throughout the entire optimization process, with errors consistently maintained at a lower level. In contrast, AMSO exhibits initial oscillations in errors, indicating that our surrogate model remains stable. While our final error may be slightly higher than that of ASMO, we believe that in cases where the errors are relatively close, the reduction in the number of optimization iterations is a highlight of our method, resulting in resource savings.

5).The description of CAND is described in Algorithm 1:

[Figure]

Figure 1: local model surrogate model cross-validation

Where, the $\Omega$ represents the random samples generated in this iteration process. $S(x)$ represents the predict value of point $x$ generated by surrogate model. $\Delta(x)$ is the distance from point $x$ to the current sampling point set $A$ and $y$ represents each point in set $A$.

We will add these to the revised manuscript.

6).We will move the cross validation to section method according to your comment.

2. The manuscript lacks a thorough mechanism analysis of how parameters affect precipitation on a global and regional scale. While section 4 presents optimization results, it lacks organization and falls short in providing a detailed understanding of the underlying mechanisms. To enhance the manuscript, it is recommended to delve deeper into the analysis. By investigating the cause-effect relationships between parameters and precipitation patterns, physics insights can be gained to improve the parameterization scheme.

1).The purpose of this work is to improve the CAM5 precipitation simulation result accord to parameter tuning method. rather than analyzing the mechanism. We believe that the goal has been achieved and it is a complete work. In this paper, we propose a surrogate model based method which can quickly calibrate parameters, and improve CAM5 precipitation using the multi-level surrogate model method and non-uniform parameterization schemes. We find a more suitable set of parameters for each region.

2).We do not change the physical processes in the parameterization scheme, we only changed the values of the parameters, or different values in different regions. We believe that these positive improvements achieved by changing the values of the parameters. In [4], there is more introduction to the mechanism, We also refer to this work when selecting parameters and determining the range of these parameters. We will try to explain these effects from the parameter value changes.

3. In equations 10-11, it could be possible for the numerator to be very large, and the denominator can be very small. This implies that the value of sigma could exceed 0.75, but the fitness is bad. If the fitness is good, the value of sigma could be close to 1 rather than just being greater than 0.75.

In order to confirm the radius of the trust region, we some works about trust region [5, 6], the update parameter $\eta 1, \eta 2$ are both less than 1 and they satisfy $0 < \eta 1 < \eta 2 < 1$. In this paper we set $\eta 1 = 0.25$ and $\eta 2 = 0.75$. If the $\sigma > 0.75$, we consider "increase the radius if the change is very successful , $\sigma \geq \eta 2$ "[7]. In our method, the surrogate model ensures a certain level of accuracy, preventing scenarios where the numerator significantly outweighs the denominator.

4. Improving the clarity of motivation for the nonuniform parameter parameterization scheme.

In this paper, the motivation for the nonuniform parameter parameterization scheme is as follows:

1).It is well known that CAM5 is a well tuned model, however globally optimal parameters do not

**Algorithm 1** Candidate point strategy
* * *
1: Compute $s^{max} \leftarrow \max_{x \in \Omega} s(x)$ and $s^{min} \leftarrow \min_{x \in \Omega} s(s)$

2: **for** each $x \in \Omega$ **do**

3:    $V^S(x) = \begin{cases} \frac{s(x) - s^{min}}{s^{max} - s^{min}} & if \ s^{max} > s^{min} \\ 1 & else \end{cases}$

4:    Calculate corresponding value of objective function for each sample.

5: **end for**

6: **for** each $x \in \Omega$ **do**

7:    $\Delta(x) = min_{y \in A} d(x, y);$

8: **end for**

9: Compute $\Delta^{max} \leftarrow \max_{x \in \Omega} s(x)$ and $\Delta^{min} \leftarrow \min_{x \in \Omega} s(x)$

10: **for** each $x \in \Omega$ **do**

11:    $V^D(x) = \begin{cases} \frac{\Delta(x) - \Delta^{min}}{\Delta^{max} - \Delta^{min}} & if \ \Delta^{max} > \Delta^{min} \\ 1 & else \end{cases}$

12: **end for**

13: **return** $argmin_{x \in \Omega} w V^S(x) + (1 - w) V^D(x)$
* * *
necessarily mean they are the best solutions for every region.

2).Regional optimization experiments demonstrate that some regions have optimal parameters, leading to better results than default parameters.

3).Our experiments show that there is a "rocker effect" in the influence of parameters on precipitation. The same parameter values have different effects on different regions, making it challenging to optimize precipitation for all regions using a single parameter.

In summary, we proposed the nonuniform parameter parameterization scheme.

5. Line 55, while previous methods involved running the climate model, it is important to note that this work also requires running the climate model in each iteration. However, the manuscript does not provide a direct comparison of the efficiency of this method with other approaches. To enhance the evaluation of the proposed method, it would be beneficial to include an assessment of the computational cost compared to existing methods. This evaluation can provide valuable insights into the efficiency and computational advantages of the proposed approach, strengthening the manuscript's contribution in terms of computational performance.

We are very willing to conduct some performance-related comparisons. However, for some commonly used parameter optimization algorithms, such as DE, PSO, GA, and so on, using these methods for parameter tuning in CAM can yield relatively good results. Nevertheless, these algorithms require more computational resources and time during execution, which makes it challenging to evaluate these methods based on performance.

Considering computational resources and time costs, we compared our method with the AMSO algorithm. The ultimate advantage in performance is the advantage in the number of iterations.

**2 Replies to minor comments**

Replies to minor issues are as follows:

1. The title uses CAM5, but the contexts use CAM. They could be consistent.

We will use consistent definition of CAM5 and other technical terms over the whole manuscript.

2. Line 11: "selected points.." to "selected points."

We will correct it in revised manuscript.

1. Line 29: traditional tuning methods in climate modeling have certain limitations. However, they remain highly useful. The majority of climate models employ traditional tuning approaches due to their reliance on well-established physics knowledge. In fact, automatic tuning methods require a solid understanding of physics to enhance their efficiency.

We agree with your comment that manual parameter tuning remains necessary. This is because optimizing the parameters of atmospheric models requires a solid understanding of the underlying physics. Our proposed method is not intended to completely replace manual tuning but to enhance the efficiency of optimization. It is built on a foundation of substantial knowledge about the model. Using

automated optimization methods, we aim to improve the tuning efficiency and reduce the consumption of computational resources. Perhaps the term "less useful" is not quite accurate. We will reconsider and use a more appropriate word to express our viewpoint.

2.Line 35, The statement that "WRF physics process is simple" is not accurate. In fact, it is known to be complex and intricate.

We agree with your comment. WRF is indeed a complex model that involves many intricate physical processes. The confusion may have arisen from our choice of words. What we are trying to emphasize is not the complexity of the model but rather that WRF is geared towards local execution and short-term forecasting, which generally incurs lower resource costs for repeated runs. In contrast, CAM5 primarily focuses on global, long-term simulations, which result in longer execution times. Therefore, when it comes to optimizing parameters for CAM, it's challenging to apply methods involving many iterations. We will replace the ambiguous terms in line with your comment.

3. Line 37, The statement that "MVFSA may become infeasible for CAM tuning" may require further consideration. Fast simulated annealing, which is utilized in MVFSA, actually requires only one population to search for the next optimal parameters. The MVFSA requires thousands of steps to get a stable solution. But CAM requires a lot of computational cost for each optimization iteration. The authors should thoroughly discuss the challenges associated with MVFSA to provide a comprehensive understanding of its feasibility for CAM tuning.

The term "infeasible" does not imply that these methods cannot be used for parameter tuning of CAM, as mentioned in the comments, MVSFA requires thousands of iterations. After these iterations, a better set of parameters can be obtained. However, from an efficiency perspective, even though this method can yield improved parameters, the computational cost and time required for thousands of iterations are deemed unacceptable. Therefore, in this paper, "infeasible" not only refers to the capability for optimization but also encompasses whether better parameters can be obtained through optimization within acceptable resource costs.

4. Line 51, When the optimization process reaches convergence, further iterations do not lead to any improvement. Similarly, once the optimization algorithm has obtained a local solution, additional iterations do not result in further enhancements. The effectiveness of the algorithm is also a determining factor in this regard.

We agree this comment. Typically, in the normal operation of an algorithm, the optimal solution improves as the number of iterations increases. However, in some cases, increasing the number of iterations may not yield any better results. This can happen when the algorithm gets stuck in a local optimum, as mentioned in the comments, or when it has already converged. We will modify this sentence to make it less absolute in revised manuscript.

5. Line 58, It is confusing that 'the mathematical expression is complex and time-consuming'. Could you explain it?

This sentence contains a punctuation error. We will rewrite it in reviesd manuscript.

6. Line 59. Revise the sentence "Wang et al. . . . ; a SCM-SMA hydrologic model"

We will rewrite this sentence in revised manuscript.

7. Line 85, the authors could carefully analyze the challenge of ASMO used in atmospheric model. The method has been successfully used in WRF, CLM. what's the real challenge for atmospheric model?

Compared to WRF, CAM focuses on long-term global-scale simulations, while WRF is designed for short-term regional-scale simulations. Consequently, CAM has higher computational costs, necessitating more efficient optimization methods. CLM is primarily employed to simulate land surface processes, while CAM is predominantly used for atmospheric processes. They encompass numerous distinct physical processes and parameterization schemes, resulting in substantial differences between the models.

While ASMO has been successfully applied to models like WRF and CLM, its application to CAM remains a challenge.

8. Line 91, the above sentences discuss the tuning algorithms. The sentence "The precipitation process . . . " talk about the metrics. It would be beneficial to separate these statements into individual paragraphs.

Yes, These sentences talk about challenge of surrogate model for precipitation parameter tuning, we will separate these statements into individual paragraphs in revised manuscript.

9. Line 110, it is hard to say the nonlinearity and complexity of CAM5 are much higher than WRF.

We agree with your point. Perhaps it's not straightforward to conclude that the nonlinearity and complexity of CAM are necessarily higher than those of WRF, as both involve a significant amount of computation and complex physical processes. We will rephrase this sentence accordingly.

10. Section 2.1, describe more details of CAM5, such as horizontal resolution, vertical level, how long does CAM5 run, the sst and sea ice are used prescribed seasonal climatology.

We will add more description about CAM5 in revised manuscript.

11. Line 138: define the six main regions, giving a table including the range of latitude and longitude.

In the manuscript, the Table 1 we introduce the region selected in the study and we will add the cite of the table in revised manuscript.

12. Line 143, why not use GPCP to estimate precipitation but use ERA5.

Both ERA5 and GPCP can be used for precipitation analysis. However ERA5 provides higher-resolution data. So that we select ERA5 to estimate precipitation.

13. Line 147, "Makes" to "makes"

We will correct it in revised manuscript.

14. Line 163, is the "sampling method" is the latin hypercube sampling? How many samples do you conduct?

Yes, the "sampling method" is latin hypercube sampling. There are 60 samples we conduct. They are described in section 3.2.1.

15. Line 280, use the correct ref for GP.

We will add the correct ref for GP in revised manuscript.

16. Line 308, It is confusing that the surrogate model is built as the quadratic function. Does it use GP?

We use the GP to construct the local-level surrogate model, the "quadratic function" means that in the initial mathematical theory of trust region , a quadratic function is used to fit the real function.

17. For fig2, what is the y-axis? Is it the relative error? How calculate it?

the y-axis represents the relative error between the predict value and real value, it is calculated based on Eq.1:

$$relative\ error = \frac{|precdictvalue - realvalue|}{realvalue} \tag{1}$$

We will revise this paragraphs according to this comment.

19. Section 4.1.3 should be merged into section 4.1.2.

We will reorganize the structure according to the the comment and other comments about the structure.

20.In figure 4, it should include the obs pattern, or the difference between opt/default and observation.

We will add new results figures in revised manuscript according to this comment.

21.Line 366, it is confusing for this sentence "Therefore, we need to further ..."

In previous sentences. We illustrate that the result of global-based tuning is not significant, some regions still need to be tuned. So that we try to find the best way to use the proposed method to improve the simulation result. Perhaps our choice of words was not precise. We will use more accurate term in this sentence.

**References**

[1] X. Wu, L. Hu, L. Wang, H. Lu, and J. Zheng. Surrogate model-based precipitation tuning for cam5. *Geoscientific Model Development Discussions*, 2023:1–30, 2023.

[2] Reza Alizadeh, Janet K Allen, and Farrokh Mistree. Managing computational complexity using surrogate models: a critical review. *Research in Engineering Design*, 31:275–298, 2020.

[3] Bianca Williams and Selen Cremaschi. Selection of surrogate modeling techniques for surface approximation and surrogate-based optimization. *Chemical Engineering Research and Design*, 170:76–89, 2021.

[4] Yun Qian, Huiping Yan, Zhangshuan Hou, Gardar Johannesson, Stephen Klein, Donald Lucas, Richard Neale, Philip Rasch, Laura Swiler, John Tannahill, Hailong Wang, Minghuai Wang, and Chun Zhao. Parametric sensitivity analysis of precipitation at global and local scales in the Community Atmosphere Model CAM5. *Journal of Advances in Modeling Earth Systems*, 7(2):382–411, June 2015.

[5] Stefan M Wild, Rommel G Regis, and Christine A Shoemaker. Orbit: Optimization by radial basis function interpolation in trust-regions. *SIAM Journal on Scientific Computing*, 30(6):3197–3219, 2008.

[6] Ruobing Chen, Matt Menickelly, and Katya Scheinberg. Stochastic optimization using a trust-region method and random models. *Mathematical Programming*, 169:447–487, 2018.

[7] Kely DV Villacorta, Paulo R Oliveira, and Antoine Soubeyran. A trust-region method for unconstrained multiobjective problems with applications in satisficing processes. *Journal of Optimization Theory and Applications*, 160:865–889, 2014.

---

## Author Comment (AC2)

**Replies to Referee #2, GMD-2023-164**

Xianwei Wu, Liang Hu, Lanning Wang, Haitian Lu, and Juepeng Zheng

November 15, 2023

Thank you very much for your patient and detailed comments on our work [1]. These valuable comments are very helpful for us to improve this paper. After carefully reading all the questions, we have answered each of them and will make appropriate corrections in the revised version of our manuscript.

In this attachment, the blue paragraphs represent your comments, and the black paragraphs below are our corresponding replies.

**1 Replies to 1-5 questions**

Replies to these questions are as follows:

1. The presentation of the algorithm and techniques could be more concise. It would further benefit from clear mathematical notation and equations, a clear nomenclature, and a clearer order. For instance, RMSE is used before properly introduced. The calculation of $V\hat{s}$ and $V\hat{d}$ are hard to follow and not right away clear.

The description of CAND is described in Algorithm 1:
* * *
**Algorithm 1** Candidate point strategy
* * *
1: Compute $s^{max} \leftarrow \max_{x \in \Omega} s(x)$ and $s^{min} \leftarrow \min_{x \in \Omega} s(s)$

2: **for** each $x \in \Omega$ **do**

3:    $V^S(x) = \begin{cases} \frac{s(x) - s^{min}}{s^{max} - s^{min}} & if \ s^{max} > s^{min} \\ 1 & else \end{cases}$

4:    Calculate corresponding value of objective function for each sample.

5: **end for**

6: **for** each $x \in \Omega$ **do**

7:    $\Delta(x) = min_{y \in A} d(x, y);$

8: **end for**

9: Compute $\Delta^{max} \leftarrow \max_{x \in \Omega} s(x)$ and $\Delta^{min} \leftarrow \min_{x \in \Omega} s(x)$

10: **for** each $x \in \Omega$ **do**

11:    $V^D(x) = \begin{cases} \frac{\Delta(x) - \Delta^{min}}{\Delta^{max} - \Delta^{min}} & if \ \Delta^{max} > \Delta^{min} \\ 1 & else \end{cases}$

12: **end for**

13: **return** $argmin_{x \in \Omega} w V^S(x) + (1 - w) V^D(x)$
* * *
Where, the $\Omega$ represents the random samples generated in this iteration process. $S(x)$ represents the predict value of point $x$ generated by surrogate model. $\Delta(x)$ is the distance from point $x$ to the current sampling point set $A$ and $y$ represents each point in set $A$.

We will add these to the revised manuscript.

We will improve the structure of the study in revised manuscript.

2. The paper lacks describing links between the physics and the choice of parameters. Why do certain parameter combinations perform better (for instance, what do they affect, how does that affect the general performance etc.)

1).The purpose of this work is to improve the CAM5 precipitation simulation result accord to parameter tuning method. rather than analyzing the mechanism. We believe that the goal has been achieved and it is a complete work. In this paper, we propose a surrogate model based method which

can quickly calibrate parameters, and improve CAM5 precipitation using the multi-level surrogate model method and non-uniform parameterization schemes. We find a more suitable set of parameters for each region.

2).We do not change the physical processes in the parameterization scheme, we only changed the values of the parameters, or different values in different regions. We believe that these positive improvements achieved by changing the values of the parameters. In [2], there is more introduction to the mechanism, We also refer to this work when selecting parameters and determining the range of these parameters. We will try to explain these effects from the parameter value changes.

3. The paper does not discuss that in general tuning for a single metric is not required as a climate model has many different metrics that need to be fulfilled. Thus, it is required to discuss how the precipitation tuning might degrade other fields. For instance, what is the effect on the global mean temperature from this tuning etc.

In this paper, we propose a parameter tuning method to improve CAM5 precipitation simulation results. The proposed method belongs to a single objective optimization method. This method maybe cannot optimize multiple objectives simultaneously. Multi-objective surrogate model-based parameter tuning method is also one of our future research topics. We will try to discuss that in nonuniform parameter parameterization scheme, how other metrics such as temperature and specific humidity change after precipitation decreases and add these results in revised manuscript.

4. The presentation of introducing the non-uniform parameter values is not entirely clear. Why should that be? What is the physical explanation for using different parameter values in different places? Shouldn't the physics be independent of the location particularly in regions which are relatively similar (South Pacific, Nino?)? Particularly you tune for different ocean regions; it is not clear why different ocean areas should have different parameter tunings. (I could understand a land vs ocean parameter change, however, different oceans or land masses requires more careful introduction and physical justification)

In this paper, the motivation for the nonuniform parameter parameterization scheme is as follows:

1).It is well known that CAM5 is a well tuned model, however globally optimal parameters do not necessarily mean they are the best solutions for every region.

2).Regional optimization experiments demonstrate that some regions have optimal parameters, leading to better results than default parameters.

3).Our experiments show that there is a "rocker effect" in the influence of parameters on precipitation. The same parameter values have different effects on different regions, making it challenging to optimize precipitation for all regions using a single parameter.

In summary, we proposed the nonuniform parameter parameterization scheme.

In section 4.3, we found that these regions have better parameter combinations, even if they are in similar positions.

5. The presentation is unclear as to why is a GP only used for the regional-level surrogate models and not for the global?

Unlike global-level surrogate-model, the local-level surrogate model is required to have a higher accuracy with fewer samples. So that construction method of the loacl model is more important. according to [3, 4]. GP is selected to construct local-level surrogate.

Similar to the global-level surrogate model, we conduct a cross-validation for several method to construct local surrogate model, the results are shown in Figure 1:

We will add these results to the rivised manuscript.

**2 Replies to major comments**

Replies to major comments are as follows:

1. LL.37-70: it would be worth discussing also the tuning approaches by Hourdin and Williamson in more detail
(http://link.springer.com/10.1007/s00382-013-1896-4 ;
http://link.springer.com/10.1007/s00382-014-2378-z ;
https://gmd.copernicus.org/articles/10/1789/2017/ ;
https://agupubs.onlinelibrary.wiley.com/doi/10.1029/2020MS002423 ;
https://onlinelibrary.wiley.com/doi/10.1029/2020MS002225 ;
https://onlinelibrary.wiley.com/doi/10.1029/2020MS002217 )

[Figure]

Figure 1: local model surrogate model cross-validation

Thank you for your suggestion. These works have many highlights in the description of the methods and the explanation of the physical mechanisms. For example, In [5, 6], authors discuss machine learning for ESM calibration and use machine learning method on both single-column model and global model. In [7, 8], authors propose a history matching method to analyse the uncertainty of parameters in HadCM3. The author also proposed the concept of "over-tuning" to prevent over fitting in the tuning results.

We believe that our work is also an extension of these tasks and a supplement to certain aspects. They are very helpful in improving our manuscript.

2.Ll.135-138: How do you inform the parameter range of the parameters to tune for? How do you choose exactly those parameters? Generally, there are more parameters in the parameterizations; why exactly those 6?

We know that the sensitivity of parameters has an important impact on pattern tuning. If insensitive parameters are selected for disturbance and adjustment, the simulation results will hardly change significantly. Therefore, the selection of sensitive parameters is one of the important conditions for CAM5 parameter tuning. Many previous studies have conducted sensitivity-related studies on precipitation-related parameters in CAM5. The parameter range and the reason we choose these parameters are according to these studies [2, 9]. These parameters we selected are proved most to sensitive precipitation in these studies.

3. LL. 140-141: Why do you choose particularly those regions? Why are they important?

These regions and there range are selected according to [10]. These regions have a high amount of precipitation value, and tuning these regions can effectively improve the simulation of CAM5 precipitation. More than that, as can be seen in Figure 2, in default experiment, these areas have a certain degree of error compared to observational data. Since CAM5 is a well-calibrated model. Our experiment also proves that the global-oriented tuning effect is not significant. So that we try to research these regions, find a set of parameters to improve the simulation results over these regions.

4. Section 3: I would suggest sticking to terminology! Whenever you talk about an actual model simulation I would suggest using ESM/CAM5 or something like that. Otherwise you start mixing up terms such as global-model, global-level, complex model which does not make it easy to follow which of all the models you refer to or whether it is a new one.

In this paper "global-model" and "global-level" model are equivalent. They represent the global-level surrogate model in the proposed method. "Complex model" represents the optimization problem which the fitness function is hard to calculate. Perhaps the word "model" has different meanings is different terms such as "surrogate model", "complex model" and "earth system model". In order to avoid

[Figure]

Figure 2: The precipitation distribution of default experiment, there are default experiment, observation data, difference between default experiment and observation data from top to bottom.

ambiguity, "complex model" can be replaced as "optimization problem", "global-level model","global model" can be replaced as "global surrogate model". We will revise these terms.

5. L 213: Why do you use reanalysis data? Why not GPCP for instance?

Both ERA5 and GPCP can be used for precipitation analysis. They both provide global precipitation data. However ERA5 provides higher-resolution data. Data of GPCP are provided on a 2.5 degree grid and ERA5 precipitation data are provided on a 0.25 degree grid. We believe that choosing data with higher resolution can significantly contrast the tuning results, thereby demonstrating the effectiveness of the proposed method. So that we select ERA5 instead of GPCP as the metric for precipitation parameter tuning. The RMSE is calculated between the CAM5 simulation results and ERA5 reanalysis data.

6. Equation 1: Comparing CAM simulations with reanalysis requires regridding data. What technique do you use for that?

The CAM5 case we used in this paper is spectral element dynamical core (SE-dycore) formulation. It need be regridded to compare with reanalysis data. We know that the ERA5 reanalysis data is lat/lon grid. We regrid the simulation result to lat/lon grid. NCL (The NCAR Command Language) is a effective tool to handle CESM simulation results data. So that we use NCL to regrid CAM simulation data to lat/lon grid. Regridding function "ESMF_regrid" is select to complete the regrid operation and "bilinear" is used for the regridding interpolation method as the function input parameter.

7. L. 249: What is the level of the fitness function? Equations for $V\hat{s}$ and $V\hat{d}$ would be very helpful!

"The level of the fitness function" means the quality of the samples, in this paper, it means the fitness values (RMSE values) of these samples. The process of CAND is shown in Section 1 question 1. It can be seen that $V^s$ represents the exploitation mechanism, it describes the simulation result of the sample $\Omega$ obtained by the current surrogate model. $V^d$ represents the exploration mechanism, it expresses the exploration for the unknown region of the current surrogate model. $V^s$ and $V^d$ is used to balance exploitation and exploration. We will add the description of CAND to the revised manuscript.

8. L. 267: "..., allowing us to estimate uncertainty from the weight parameter": How do you estimate uncertainty of the weight parameter?

Perhaps there is some ambiguity in our expression, in this sentence, the weight parameter represents $V^D$ in CAND strategy, we use the parameter $V^D$ to represent the exploration mechanism, which means searching a new optimum in uncharted regions.

The sentence does not means that we want to estimate the uncertainty of the parameters on the CAM5 simulation results.

Maybe these terms like "uncertainty", "weight parameter" have other meanings in ESM parameter tuning. We revise the sentence as follows:

"The set of the previously sampled points denote the region which has been explored, allowing us to estimate the uncertainty from the distance between the generated point sets to the explored region."

Thanks for your comment. The two sentences both describe that add new samples to update surrogate model. We delete the sentences with similar meanings. The new sentences are as follows:

"Generally, when solving a complex parameter optimization problem by a surrogate model, to improve the accuracy simulation results of the surrogate model, additional new sample points need to be added. Thus reducing the number of simulations of the actual complex model."

We will add these sentences in revised manuscript.

10. Figure 1: Could you put the whole algorithm into the flowchart which indicates which technique is used at which step which would make the whole description much clearer.

Thanks for your comment, We will update the flowchart according to your comment, in each step we will add which technique is used. The new flowchart are shown in Figure 3.

We believe that the new flowchart is much clearer. Compared with the old flowchart, we add the whole process of CAND and trust region method. In order to show the integration of these process, dotted bordered rectangle is used to mark the method which the current process belongs to. In addition, we add the surrogate construction method of each level in the new flowchart.

We will add the new figure in revised manuscript.

11. Figure 3: Why is the relative error of the proposed method increasing with iterations? Shouldn't the surrogate model get more accurate with iteration numbers?

Ideally, the error will gradually decrease, but there will be some inevitable small oscillations during the optimization process, resulting in a slight increase in error.

[Figure]

Figure 3: New flowchart

it's possible that our method may indeed have slightly higher errors compared to ASMO in the end. However, our method demonstrates greater stability throughout the entire optimization process, with errors consistently maintained at a lower level. In contrast, AMSO exhibits initial oscillations in errors, indicating that our surrogate model remains stable. While our final error may be slightly higher than that of ASMO, we believe that in cases where the errors are relatively close, the reduction in the number of optimization iterations is a highlight of our method.

12. L 256: "...we only run the real model once, and": But don't you add more than one sample in each iteration? So, how do you need ot run the model only once?

We run the real model (CAM) once in each iteration step, and add the parameters and corresponding RMSE value to sampling set. For example, if the whole tuning process contains 30 steps, in each step one pair of parameters and RMSE value are added to sampling set. There are 30 pairs added to sampling set. We can rewrite the sentence as:

"For the whole tuning process, we only run the CAM5 model once in each iteration step, and all of the samples are predicted using the surrogate model."

We believe that new sentences will express their meanings more clearly, and we will add them in revised manuscript.

13. LL. 377-384: "I think it is important to understand how each parameter influences the model simulations. Why did you pick only one here? This section requires more careful exploration of the parameter itself and the physical mechanisms. What are the physical reasons for the positive and negative correlations described in ll. 377-379

We agree your comment that it is important to understand how each parameter influences the model simulations. The motivation of this work is to propose a tuning method and how to use the mothod for CAM5 precipitation tuning. We analyse the parameter rhminl to prove the "rocker effect": The values of parameters will have different impacts in different regions. Thus introduce a more appropriate way to use the surrogate-based method and the nonuniform parameter parameterization scheme. We are willing to study how each parameter influences the model simulations in future research.

14. Table3/4: Could you also discuss the RMSE of those regions from the optimized parameter set

Table 1: The CAM simulation performance increase for each region.

| Region | Default RMSE | Global surrogate RMSE | Optimized RMSE | Reduction Rate |
|--------|--------------|------------------------|-----------------|-----------------|
| WarmPool | 1.985 | 1.961 | 1.924 | 3.07% |
| South Pacific | 0.855 | 0.788 | 0.455 | 46.78% |
| Niño | 0.931 | 0.855 | 0.773 | 17.04% |
| South America | 2.576 | 2.459 | 2.371 | 7.94% |
| South Asia | 1.484 | 1.352 | 1.293 | 12.87% |
| East Asia | 1.213 | 1.043 | 0.878 | 27.68% |

from the global-surrogate model? This would clarify what the gain is from the local-level surrogate models.

We agree your comment, discuss the improve from different level of surrogate model will better demonstrate the effectiveness of our method. We will add the tuning result obtained from global-level surrogate. The new table are shown in Table 2

The "Global surrogate RMSE" means the optimization result obtained by global-level surrogate and "Optimized RMSE" represents the final result.

Table 4 in manuscript represents the results of nonuniform parameter parameterization scheme, which does not involve any level surrogate model or new tuning process.

**3 Replies to minor comments**

Replies to minor comments are as follows:

1. L. 1: "The uncertainty of physical parameters is a major reason for a poor precipitation simulation performance in Earth system models (ESMs), especially over the tropical and Pacific regions.": Is it not only uncertainty of physical parameters but also the microphysics parameterizations itself.

We agree your comment, microphysics parameterization is also one of the main factors influencing the precipitation. Our wording might be too absolute. We should use words like "one of the major reasons".

2. Ll. 14-15: "The results show that the surrogate model-based optimization method can significantly improve the simulation performance of the CAM model.": I would rephrase it stating that the surrogate model-based optimization method allows for better identifying optimal parameter values.

We agree your comment, we will revise the sentence according to your comment.

3. L. 25: "...could lead to huge deviations in the simulations": Deviations from what?

We know that some parameters are sensitive for the simulation result value (eg. temperature, precipitation). A slight change in these parameter values can lead to significant numerical variations in the simulation results. Perhaps our choice of words was not precise. "Error" might be a more accurate term in this sentence.

4. LL. 168-169: "The strategy leverages the information and knowledge obtained from the surrogate model to optimize the run time of the real complex model to fulfill the requirement of accuracy.": How do you optimize for the run time of the real complex model (I guess the ESM)?

The real complex model is the ESM in this paper, we will revise these terms to avoid ambiguity. The appropriate strategy can reduce the number of iteration so as to reduce run time of the real complex model. Perhaps our choice of words was not precise. We will use more accurate term in this sentence.

5. L 171: ".. to update the global-level surrogate model until global-model convergence.": Which global-model do you refer to here, which global-model has to converge?

The global model is the global-level surrogate model. Only when the global-level surrogate model converges, the local-level surrogate model will be created. We will revise these terms to avoid ambiguity.

6. L.172: "... high-waulity CAM" Do you mwan with high-waulity simulations closer to the target value?

Yes, "the high-quality results" means the simulations closer to the target value, in this paper, is the reanalysis data.

7. L. 175-176: "In the parameter tuning process, each surrogate model can fully explore the parameter space to obtain better solutions, generating a large number of samples.": I don't understand

this sentence as earlier (ll 171-172) it is stated that local-level surrogates don't use the whole parameter space?

"Parameter space" means the range of all the parameters, local-level surrogate don't use the whole samples, only high quality samples selected. However the parameter searching in tuning process of local-level surrogate is also over the range of all the parameters (parameter space).

8. LL. 202-211: Why do you talk about 1-D LHS. Usually LHS code can handle several dimensions. LHS code usually makes sure to maximize the minimal distance between all vectors in order to sample the whole space as uniformly as possible.

We agree your comment that LHS code can handle several dimensions. We use a LHS to generate samples, there are 6 parameters selected in this paper. Introduction of 1-D LHS is a example to describe the process of LHS. We use 6-d LHS in this paper.

9. Ll. 222-225: The two sentences appear to have very similar information and should be rewritten.

We will rewrite these sentences in revised manuscript.

10. ll. : 347: Do you compare here the RMSEs of the final optimized parameter set? If so, do they converge to the same parameter set or different ones?

Yes, we compare RMSE of the simulation results of different parameters. However, they converge to different parameter values.

11. Figure 4: It would be good to see what you are tuning for. Can you also add the target value and not only the default and the biases to the target?

We will add these figures according to your comment.

12. L 370: What do you mean with influence mode?

The "influence mode" means the impact of changes in parameter values on the precipitation of each grid, positive correlation or negative correlation.

**4 Replies for technical issues**

Replies for technical issues are as follows:

1. Labels on contour plots are generally very small.

We will revise these plots according to this comment.

2. L. 63: I might have missed it but "ANNs" acronym was not introduced.

Acronyms will be properly defined in our revised manuscript.

3. L. 78: I might have missed it but "SCA-SMA" acronym was not introduced.

Acronyms will be properly defined in our revised manuscript.

4. L.132: "The compset used in this study is F_2000_CAM5, and the resolution is ne30_g16": To the normal reader these abbreviations don't mean anything. A little bit more explanation would be nice. What is F_2000_CAM5 for instance or what does ne30_g16 mean in the physical world?

We will add more description about CAM5 and the compset used in this study in revised manuscript.

5. Can you use maybe mathematical notation of the original parameters in the paraemeterization instead of the CAM5 parameter naming? For instance line. 148 zmconv_tau is simply tau in the original Zhang McFarlane paper.

We will use mathematical notation of the parameters in revised manuscript.

6. L 275: "actual situation": you probably mean behaviour or something like that.

We agree your comment, it may be more accurate if we use "behaviour" to replace to the "situation".

7. Figure 2: y-axis label missing

We will correct it in revised manuscript.

8. L. 333: What is now X,Y in the S{X,Y} notation?

{X,Y} represents the pairs of parameters and corresponding RMSE value. X represents the parameter combinations of selected 6 parameters and Y represents the RMSE values obtained from CAM simulation results of corresponding parameters.

9. Figure 3: no x-axis label

We will correct it in revised manuscript.

10. L. 398: "precipitation change trend": What trend do you mean here? Time trend?

The "trend" means that South Pacific region generally reveals a ladder-like decline from east to west, which is described in last sentence.

**References**

[1] X. Wu, L. Hu, L. Wang, H. Lu, and J. Zheng. Surrogate model-based precipitation tuning for cam5. *Geoscientific Model Development Discussions*, 2023:1–30, 2023.

[2] Yun Qian, Huiping Yan, Zhangshuan Hou, Gardar Johannesson, Stephen Klein, Donald Lucas, Richard Neale, Philip Rasch, Laura Swiler, John Tannahill, Hailong Wang, Minghuai Wang, and Chun Zhao. Parametric sensitivity analysis of precipitation at global and local scales in the Community Atmosphere Model CAM5. *Journal of Advances in Modeling Earth Systems*, 7(2):382–411, June 2015.

[3] Reza Alizadeh, Janet K Allen, and Farrokh Mistree. Managing computational complexity using surrogate models: a critical review. *Research in Engineering Design*, 31:275–298, 2020.

[4] Bianca Williams and Selen Cremaschi. Selection of surrogate modeling techniques for surface approximation and surrogate-based optimization. *Chemical Engineering Research and Design*, 170:76–89, 2021.

[5] Frédéric Hourdin, Daniel Williamson, Catherine Rio, Fleur Couvreux, Romain Roehrig, Najda Villefranque, Ionela Musat, Laurent Fairhead, F Binta Diallo, and Victoria Volodina. Process-based climate model development harnessing machine learning: Ii. model calibration from single column to global. *Journal of Advances in Modeling Earth Systems*, 13(6):e2020MS002225, 2021.

[6] Fleur Couvreux, Frédéric Hourdin, Daniel Williamson, Romain Roehrig, Victoria Volodina, Najda Villefranque, Catherine Rio, Olivier Audouin, James Salter, Eric Bazile, et al. Process-based climate model development harnessing machine learning: I. a calibration tool for parameterization improvement. *Journal of Advances in Modeling Earth Systems*, 13(3):e2020MS002217, 2021.

[7] Daniel Williamson, Adam T Blaker, Charlotte Hampton, and James Salter. Identifying and removing structural biases in climate models with history matching. *Climate dynamics*, 45:1299–1324, 2015.

[8] Daniel Williamson, Michael Goldstein, Lesley Allison, Adam Blaker, Peter Challenor, Laura Jackson, and Kuniko Yamazaki. History matching for exploring and reducing climate model parameter space using observations and a large perturbed physics ensemble. *Climate dynamics*, 41:1703–1729, 2013.

[9] Raju Pathak, Sandeep Sahany, and Saroj K. Mishra. Uncertainty quantification based cloud parameterization sensitivity analysis in the NCAR community atmosphere model. *Scientific Reports*, 10(1):17499, December 2020.

[10] Ben Yang, Yun Qian, Guang Lin, L. Ruby Leung, Philip J. Rasch, Guang J. Zhang, Sally A. McFarlane, Chun Zhao, Yaocun Zhang, Hailong Wang, Minghuai Wang, and Xiaohong Liu. Uncertainty quantification and parameter tuning in the CAM5 Zhang-McFarlane convection scheme and impact of improved convection on the global circulation and climate. *Journal of Geophysical Research: Atmospheres*, 118(2):395–415, 2013. _eprint: https://agupubs.onlinelibrary.wiley.com/doi/pdf/10.1029/2012JD018213.

---

## Author Comment (AC3)

**Replies to Referee #1, GMD-2023-164**

Xianwei Wu, Liang Hu, Lanning Wang, Haitian Lu, and Juepeng Zheng

November 30, 2023

Thank you very much for your patient and detailed comments on our work [1]. These valuable comments are very helpful for us to improve this paper. After carefully reading all the questions, we have answered each of them and will make appropriate corrections in the revised version of our manuscript.

In this attachment, the blue paragraphs represent your comments, and the black paragraphs below are our corresponding replies.

**1 Replies to major comments**

Replies to major comments are as follows:

1. The manuscript structure, particularly the method section, needs to be reorganized to improve the compactness. There are several areas that require clarification. For instance, Algorithm 1 calculates the RMSE, but its definition is found in section 3.2.2. It would be more appropriate to move the definition to section 2. Additionally, in Line 4 of Algorithm 2, it is unclear whether the new parameters are obtained using CAND. Furthermore, it is not explained why the local-level surrogate utilizes Gaussian Process. In addition, it could describes the difference between the algithm used in this work and the ASMO. Typically, optimization algorithms require hundreds of steps to achieve convergence, but in this work, only around 20 steps of local optimization are performed. It is hard to say the algorithms get convergence. It appears that the ASMO method can achieve local optimization more quickly. The conclusion is not convinced. The description of CAND is difficult to follow, particularly the calculation vs and vd, which is lack of calculation details. The cross validation describe can move from result section to the method section.

Thanks for your comment. This comment contains multiple questions, we will reply these questions separately.

1) We will improve the structure of the study in revised manuscript according to the comments.

2) The new parameters are not obtained by CAND, CAND is just used for global-level surrogate model. $x_t$ is the optimal solution within the trust region of the surrogate model. $f(x_t)$ represents the objective function (in this problem is RMSE) of the simulation results of this parameter set in CAM5.

3) In the process of constructing the global surrogate, because of the relatively large amount of samples, we chose some relatively complex learning-based models and selected the optimal method based on cross-validation results. The results in this paper indicates that the GBRT is the best method to construct global surrogate model.

In the process of building local surrogate models, we take into account the insufficient number of samples. Using some relatively complex learning-based methods may lead to under-fitting. Therefore, we consider selecting some relatively simple construction methods. Among them, Polynomial Response Surface (PRS), Gaussian Process (GP), and Radial Basis Function (RBF) are the most commonly used regression-based or statistics-based methods, widely used for various complex parameter optimization

[Figure]

Figure 1: Local surrogate model cross-validation results

problems in the industry [2]. Therefore, we consider choosing one of these three as the method for constructing local surrogate models. Among them, we first consider PRS. Although it is the simplest method, the fitting performance of the PRS model to complex curves is relatively poor. PRS model is a simple model based on polynomials, and it may not perform well for complex, nonlinear, or highly interactive systems. Its expressive capacity is limited and may not accurately capture certain complex relationships. According to [3], the polynomial surrogate model does not perform well in terms of fitting accuracy for multivariate and nonlinear problems. Therefore, we choose either GP or RBF to construct the local surrogate model.

In order to choose a model that is more suitable for our study, we conducted cross-validation experiments based on the selection method for the global model. We selected three learning-based methods: Random Forest (RF), Support Vector Machine (SVM), and Artificial Neural Network (ANN) for comparison. The results are shown in the figure 1. The results indicate that compared with RBF method, GP has a smaller error in cross-validation, providing more accurate predictions. Moreover, the cross-validation results are better than the three learning-based methods. In contrast, the RBF method not only has a larger error but also a wider range of upper and lower relative error bounds. The prediction results are unstable, and the predictive performance is lower than the three learning-based surrogate model construction methods.

**Please note that in these paragraphs, "global" and "local" represent the different surrogate model, rather than "global/local optimal" in optmization process or simulation results in "global/region".**

We will add the relevant experimental results to the revised manuscript.

4) The entire optimization process consists of over 20 steps. After obtaining the current optimal solution, there are several validation steps. Once these validation steps are completed, and no new optimal solution is found, we consider the current optimal solution as the final result of the optimization. In the figure, we illustrate the reduction in RMSE during the optimization process and the associated errors.

In terms of errors, it's possible that our method may indeed have slightly higher errors compared

to ASMO in the end. However, our method demonstrates greater stability throughout the entire optimization process, with errors consistently maintained at a lower level. In contrast, AMSO exhibits initial oscillations in errors, indicating that our surrogate model remains stable. While our final error may be slightly higher than that of ASMO, we believe that in cases where the errors are relatively close, the reduction in the number of optimization iterations is a highlight of our method, resulting in resource savings.

5) CAND (candidate points) strategy is proposed in [4], has been widely used in surrogate model-based optimization method [5, 6]. It is used for balance the global exploration and local exploitation. **Please note that the terms "global and local" represent characters of the optimal solution in optimization methods rather than "best parameter for global precipitation or region precipitation".**The inputs of the strategy are a set of generated points $\Omega$, a set of initial sampling points $A$, and the output of the strategy is the best candidate point selected from the generated points $\Omega$.

In each iteration step, there are two criteria in CAND to select to select the best candidate point:

i). Estimated function value obtained from the surrogate model.

ii). Minimum distance from previously evaluated points.

The first criterion represents the exploitation, which means that search a better solution based on known regions. The second criterion represents the exploration, which means that search a better solution in an unknown region. **Please note that in this sentence "region" represents a part of parameter space rather than precipitation simulation results over each "region" like East Asia.** In order to find the next candidate for evaluation, we do not minimize that fitness function over a continuous set. Instead, we select the best among a finite set of randomly generated points. Fitness functions are made up from these two criteria: the value of fitness function obtained from the surrogate model at each point and its minimum distance to existing data points.

The description of CAND is described in Algorithm 1:
* * *
**Algorithm 1** Candidate point strategy
* * *
1: Compute $s^{max} \leftarrow \max_{x \in \Omega} s(x)$ and $s^{min} \leftarrow \min_{x \in \Omega} s(s)$
2: **for** each $x \in \Omega$ **do**
3: $\quad V^S(x) = \begin{cases} \frac{s(x) - s^{min}}{s^{max} - s^{min}} & if \ s^{max} > s^{min} \\ 1 & else \end{cases}$
4: $\quad$ Calculate corresponding value of objective function for each sample.
5: **end for**
6: **for** each $x \in \Omega$ **do**
7: $\quad \Delta(x) = min_{y \in A} d(x, y);$
8: **end for**
9: Compute $\Delta^{max} \leftarrow \max_{x \in \Omega} s(x)$ and $\Delta^{min} \leftarrow \min_{x \in \Omega} s(x)$
10: **for** each $x \in \Omega$ **do**
11: $\quad V^D(x) = \begin{cases} \frac{\Delta(x) - \Delta^{min}}{\Delta^{max} - \Delta^{min}} & if \ \Delta^{max} > \Delta^{min} \\ 1 & else \end{cases}$
12: **end for**
13: **return** $argmin_{x \in \Omega} w V^S(x) + (1 - w) V^D(x)$
* * *
Where, the $\Omega$ represents the random samples generated in this iteration process. $S(x)$ represents the predict value of point $x$ generated by surrogate model. $\Delta(x)$ is the minimum distance from point $x$ to the current sampling point set $A$ and $y$ represents each point in set $A$. In line 1, fitness function values of generated point sets are calculated according to the surrogate model and their maximum and minimum are marked. In line 2-4, the value of $V^S$ is calculated and the for loops represents the

criterion 1, a smaller value of $V^S$ means that the current point is an effective exploitation. In line 6-8, the minimum distance $\Delta(x)$ between the point sample $\Omega$ and current sampling point set is calculated in the for loops. The maximum and minimum of $\Delta(x)$ are marked in line 9. The for loops in line 10-12 represent the criterion 2, a smaller value of $V^D$ means that the current point is an effective exploration. The last line means the weighted sum to balance the exploration and exploitation.

We will add these to the revised manuscript in section 3.2.3.

6) We will move the cross validation to section method according to your comment.

2. The manuscript lacks a thorough mechanism analysis of how parameters affect precipitation on a global and regional scale. While section 4 presents optimization results, it lacks organization and falls short in providing a detailed understanding of the underlying mechanisms. To enhance the manuscript, it is recommended to delve deeper into the analysis. By investigating the cause-effect relationships between parameters and precipitation patterns, physics insights can be gained to improve the parameterization scheme.

Thanks for your comment. We try to reply to this comment from the following points.

1).The purpose of this work is to improve the CAM5 precipitation simulation result accord to parameter tuning method, rather than analyzing the mechanism of physics process. We believe that the goal has been achieved and it is a complete work. In general, the calibration of parameters in parameterization schemes relies on statistical models and expert knowledge, leading to significant uncertainty. Small variations in parameter values can result in substantial changes in simulation results. In this paper, we propose a surrogate model based method which can quickly calibrate parameters, and improve CAM5 precipitation using the proposed multi-level surrogate model method. During the optimization process, we found that the same parameter value has different effects on simulation results in different regions, called "rocker effect". This suggests that it is challenging to achieve precipitation optimization for all regions using the same set of parameters. Therefore, we design a non-uniform parameterization scheme, employing different parameter values for distinct regions. We find a more suitable set of parameters for each region according to the multi-level surrogate model-based method and integrate these different sets of parameter values into one case.

2).We do not change the physical process in the parameterization schemes, we only changed the values of the parameters, or different values in different regions. In this paper, we use CAM5 and the selected parameters belong to three different parameterization schemes: The cloud microphysics parameterization scheme is proposed by [7, 8, 9]. The deep convective parameterization scheme is proposed by [10] and modified by [11]. The cloud fraction parameterization scheme is proposed by [12]. We believe that these studies have analyzed the physical meanings of each parameter and their effects on simulation results. While maintaining these physical processes, we improve the parameterization scheme based on the optimal values obtained for each region through the proposed tuning methods. In the improved parameterization scheme, the parameter values for each selected region are set to the tuned values, while the values for other regions remain at default parameters. In the parameterization scheme file, there have been no changes to the descriptions of physical processes. The modifications we made involve selecting different numerical values based on the judgment of region latitudes and longitudes. So that we believe that these positive improvements achieved by changing the values of the parameters. In [13], there is more introduction to the mechanism of physics process, including the impact of each parameter on precipitation results in different regions. We also refer to this work when selecting parameters and determining the range of these parameters. We will try to explain these effects from the parameter value changes according to this work and any other related works.

3. In equations 10-11, it could be possible for the numerator to be very large, and the denominator

[Figure]

Figure 2: Sigma value

can be very small. This implies that the value of sigma could exceed 0.75, but the fitness is bad. If the fitness is good, the value of sigma could be close to 1 rather than just being greater than 0.75.

In order to confirm the radius of the trust region, we some works about trust region [14, 15], the update parameter $\eta 1, \eta 2$ are both less than 1 and they satisfy $0 < \eta 1 < \eta 2 < 1$. In this paper we set $\eta 1 = 0.25$ and $\eta 2 = 0.75$. If the $\sigma > 0.75$, we consider "increase the radius if the change is very successful , $\sigma \geq \eta 2$ "[16]. In our method, the surrogate model ensures a certain level of accuracy, preventing scenarios where the numerator significantly outweighs the denominator.

To validate whether the value of $\sigma$ is reasonable, we extract and plot the sigma values during the optimization and validation convergence processes, as shown in Figure 2. It can be observed that all values are distributed between 0.6 and 1.6, indicating that the error of the surrogate model is generally controlled within a certain range, without exceptionally small or large outliers.

We believe that the setting of $\sigma$ is free from anomalies and can effectively optimize the precipitation parameters of CAM5.

4. Improving the clarity of motivation for the nonuniform parameter parameterization scheme.

Thanks for your comment. We try to reply to this comment from the following points.

1) In this paper, the motivation for the nonuniform parameter parameterization scheme is as follows:

- It is well known that CAM5 is a well tuned model, however the holistic optimal parameters do not necessarily mean they are the best solutions for every region. The simulation results over these regions are challenging to improve through global tuning experiments. **Please note**

[Figure]

Figure 3: Global spatial distributions of relative contributions (%) of parameters to total variance of annual mean precipitation in [13].

>that in this sentence "global" tuning represents the CAM5 tuning experiments to improve the global simulation results of precipitation rather than "global optimal" in optimization process.

- Regional optimization experiments demonstrate that some regions have optimal parameters, leading to better results than default parameters. However, applying these parameters to global simulations may not obtain optimal results; they are effective only within the selected regions for achieving the best simulation outcomes.

- Our experiments show that there is a "rocker effect" in the influence of parameters on precipitation. The same parameter values have different effects on different regions. When the simulation results in one region improve due to changes in parameter values, the results in other regions may decline. This implies that optimizing precipitation for all regions using a single set of parameters is challenging.

2) In [13], the authors discuss the contributions of different parameters to precipitation in different regions. The research results indicate that the contribution of different parameters to precipitation varies across regions. As shown in the figure 3, it can be observed that the contribution of parameters to precipitation cannot be simply judged based on the relationship with ocean or land. Even in adjacent regions, there can be some degree of differences. When using globally uniform parameter values, in order to pursue a holistic optimal solution, approximate mean value is employed to achieve a better overall simulation performance. If there is a significant difference between the local optimum and mean value in certain regions, the simulation results for that region will have a large error. **Please note that "local optimum" in this sentence means that the best parameters over this region, rather than "Local optimum solution" in the optimization process.**

3) We must also consider the diversity of the oceans regions. In CAM5, the physical processes

[Figure]

Figure 4: climatology SST

related to the ocean include optical reflection and some complex thermodynamic processes. The variations in sea surface temperatures have a significant impact on these physical processes, leading to substantial differences in simulation results across different regions. As can be seen in the Figure 4, the sea surface temperatures vary significantly across different oceanic regions, such as the Pacific region and warmpool. In the presence of such differences, dividing parameters based solely on ocean/land distinctions is not precise enough. Therefore, we try to select multiple regions and utilize faster parameter tuning method to find better parameters. These papameters are then integrated into the same case through a non-uniform parameterization scheme.

In summary, we proposed the nonuniform parameter parameterization scheme. We search different parameter combinations for different regions by surrogate model based tuning method and integrate them into a single case in a non-uniform parameterization scheme.

5. Line 55, while previous methods involved running the climate model, it is important to note that this work also requires running the climate model in each iteration. However, the manuscript does not provide a direct comparison of the efficiency of this method with other approaches. To enhance the evaluation of the proposed method, it would be beneficial to include an assessment of the computational cost compared to existing methods. This evaluation can provide valuable insights into the efficiency and computational advantages of the proposed approach, strengthening the manuscript's contribution in terms of computational performance.

Thanks four your comment. We are very willing to conduct some performance-related comparisons. However, for some commonly used parameter optimization algorithms, such as DE, PSO, GA, and so on, using these methods for parameter tuning in CAM can yield relatively good results. Nevertheless, these algorithms require more computational resources and time during execution, which makes it challenging to evaluate these methods based on performance.

We attempt to use algorithms such as PSO and GA for parameter tuning for the CWRF model (Climate-Weather Research and Forecasting model) [17]. We know that CWRF, as a weather forecasting model, has much shorter runtime and resource consumption compared to CAM5. We refer to these commonly used parameter settings, setting the population size to 50 and the number of iterations to 100. The results show that even for a model with significantly shorter execution time and computational resource consumption than CAM5, these optimization methods still struggle to obtain optimal parameters within a reasonable time frame. The results may even be insufficient to meet the

timeliness requirements for CWRF predictions. While comparing the performance advantages of our method with other methods can demonstrate the superiority of our approach, the performance of these methods is challenging to quantify in a short time frame due to limitations such as time constraints and allocated computational resources.

Considering computational resources and time costs, we compare our method with the AMSO algorithm. The ultimate advantage in performance is the advantage in the number of iterations.

**2 Replies to minor comments**

Replies to minor issues are as follows:

1. The title uses CAM5, but the contexts use CAM. They could be consistent.

Thanks for your comment. We will use consistent definition of CAM5 and other technical terms over the whole manuscript.

2. Line 11: "selected points.." to "selected points."

Thanks for your comment. We will correct it in revised manuscript.

1. Line 29: traditional tuning methods in climate modeling have certain limitations. However, they remain highly useful. The majority of climate models employ traditional tuning approaches due to their reliance on well-established physics knowledge. In fact, automatic tuning methods require a solid understanding of physics to enhance their efficiency.

Thanks for your comment. We agree with your comment that manual parameter tuning remains necessary. This is because optimizing the parameters of atmospheric models requires a solid understanding of the underlying physics. Our proposed method is not intended to completely replace manual tuning but to enhance the efficiency of optimization. It is built on a foundation of substantial knowledge about the model. Using automated optimization methods, we aim to improve the tuning efficiency and reduce the consumption of computational resources. Perhaps the term "less useful" is not quite accurate. We will reconsider and use a more appropriate word to express our viewpoint.

2.Line 35, The statement that "WRF physics process is simple" is not accurate. In fact, it is known to be complex and intricate.

Thanks for your comment. We agree with your comment. WRF is indeed a complex model that involves many intricate physical processes. The confusion may have arisen from our choice of words. What we are trying to emphasize is not the complexity of the model but rather that WRF is geared towards local execution and short-term forecasting, which generally incurs lower resource costs for repeated runs. In contrast, CAM5 primarily focuses on global, long-term simulations, which result in longer execution times. Therefore, when it comes to optimizing parameters for CAM, it's challenging to apply methods involving many iterations. We will replace the ambiguous terms in line with your comment.

3. Line 37, The statement that "MVFSA may become infeasible for CAM tuning" may require further consideration. Fast simulated annealing, which is utilized in MVFSA, actually requires only one population to search for the next optimal parameters. The MVFSA requires thousands of steps to get a stable solution. But CAM requires a lot of computational cost for each optimization iteration. The authors should thoroughly discuss the challenges associated with MVFSA to provide a comprehensive understanding of its feasibility for CAM tuning.

Thanks for your comment. The term "infeasible" does not imply that these methods cannot be used for parameter tuning of CAM, as mentioned in the comments, MVSFA requires thousands of iterations. After these iterations, a better set of parameters can be obtained. However, from an

efficiency perspective, even though this method can yield improved parameters, the computational cost and time required for thousands of iterations are deemed unacceptable. Therefore, in this paper, "infeasible" not only refers to the capability for optimization but also encompasses whether better parameters can be obtained through optimization within acceptable resource costs.

4. Line 51, When the optimization process reaches convergence, further iterations do not lead to any improvement. Similarly, once the optimization algorithm has obtained a local solution, additional iterations do not result in further enhancements. The effectiveness of the algorithm is also a determining factor in this regard.

Thanks for your comment. We agree this comment. Typically, in the normal operation of an algorithm, the optimal solution improves as the number of iterations increases. However, in some cases, increasing the number of iterations may not yield any better results. This can happen when the algorithm gets stuck in a local optimum, as mentioned in the comments, or when it has already converged. We will modify this sentence to make it less absolute in revised manuscript.

5. Line 58, It is confusing that 'the mathematical expression is complex and time-consuming'. Could you explain it?

Thanks for your comment. This sentence contains a punctuation error. We will rewrite it in reviesd manuscript.

The sentence can be rewrite as:

**The objective function of these problems is difficult to describe as a mathematical expression or the mathematical expression is complex and time-consuming.**

6. Line 59. Revise the sentence "Wang et al. ... ; a SCM-SMA hydrologic model"

Thanks for your comment. We will rewrite this sentence in revised manuscript.

The sentence can be rewrite as:

**Wang et al. (2014) established a connection between the optimization of mathematical benchmarks and complex geoscientific models, and proposed the adaptive surrogate model-based optimization (ASMO) method. In order to demonstrate the performance of ASMO method for parameter tuning ofcomplex geoscientific models, parameters of a SCA-SMA hydrologic model was tuned based on the ASMO method.**

7. Line 85, the authors could carefully analyze the challenge of ASMO used in atmospheric model. The method has been successfully used in WRF, CLM. what's the real challenge for atmospheric model?

Thanks for your comment. WRF is a regional model that focuses more on simulating specific regions, with a smaller spatial scale. The differences and variations in parameter values within a region are not particularly large. Therefore, the computational resource cost of parameter tuning is not exceptionally high. CLM is a land model designed specifically for simulating processes occurring on Earth's land. Considering that a significant portion of the Earth's surface is covered by oceans, which involve complex physical and chemical changes, as well as direct energy and substance exchanges between land and ocean, these aspects are not incorporated into CLM. In contrast, CAM includes physical processes related to the ocean, resulting in notable differences between the two models.

So that ASMO has been successfully applied to models like WRF and CLM, the application of surrogate model-based parameter tuning to CAM remains a challenge. In this paper, we designe a multi-level surrogate model-based method and demonstrate that the surrogate-based methods can enhance precipitation simulation results in CAM5.

8. Line 91, the above sentences discuss the tuning algorithms. The sentence "The precipitation process ..." talk about the metrics. It would be beneficial to separate these statements into individual

paragraphs.

Thanks for your comment. Yes, These sentences talk about challenge of surrogate model for precipitation parameter tuning, we will separate these statements into individual paragraphs in revised manuscript.

9. Line 110, it is hard to say the nonlinearity and complexity of CAM5 are much higher than WRF.

Thanks for your comment. We agree with your point. Perhaps it's not straightforward to conclude that the nonlinearity and complexity of CAM are necessarily higher than those of WRF, as both involve a significant amount of computation and complex physical processes. We will rephrase this sentence accordingly.

10. Section 2.1, describe more details of CAM5, such as horizontal resolution, vertical level, how long does CAM5 run, the sst and sea ice are used prescribed seasonal climatology.

Thanks for your comment. We will add more description about CAM5 and the compset used in this study. They include the modes used in the compset, the description of the grid and their specific meanings. We will supplement to the revised manuscript based on [18]

11. Line 138: define the six main regions, giving a table including the range of latitude and longitude.

Thanks for your comment. In the manuscript, the Table 1 we introduce the region selected in the study and we will add the cite of the table in revised manuscript.

12. Line 143, why not use GPCP to estimate precipitation but use ERA5.

Thanks for your comment. Both ERA5 and GPCP can be used for precipitation analysis. They both provide global precipitation data. However ERA5 provides higher-resolution data. Data of GPCP are provided on a 2.5 degree grid and ERA5 precipitation data are provided on a 0.25 degree grid. We believe that choosing data with higher resolution can significantly contrast the tuning results, thereby demonstrating the effectiveness of the proposed method. So that we select ERA5 instead of GPCP as the metric for precipitation parameter tuning. The RMSE is calculated between the CAM5 simulation results and ERA5 reanalysis data.

13. Line 147, "Makes" to "makes"

Thanks for your comment. We will correct it in revised manuscript.

14. Line 163, is the "sampling method" is the latin hypercube sampling? How many samples do you conduct?

Thanks for your comment. Yes, the "sampling method" is latin hypercube sampling. There are 60 samples we conduct. They are described in section 3.2.1.

15. Line 280, use the correct ref for GP.

Thanks for your comment. We will add the correct ref for GP in revised manuscript.

16. Line 308, It is confusing that the surrogate model is built as the quadratic function. Does it use GP?

Thanks for your comment. We use the GP to construct the local-level surrogate model, the "quadratic function" means that in the initial mathematical theory of trust region , a quadratic function is used to fit the real function. These sentence are used to describe the trust region theory rather than introduce our proposed method.

17. For fig2, what is the y-axis? Is it the relative error? How calculate it?

Thanks for your comment. the y-axis represents the relative error between the predict value and real value, it is calculated based on Eq.1:

$$relative\ error = \frac{|precdictvalue - realvalue|}{realvalue} \qquad (1)$$

Where, the *precdictvalue* represents the objective function value (in this paper is RMSE) predicted by surrogate model and *realvalue* represents the objective function value (in this paper is RMSE) obtained by CAM5 simulation. We will revise this paragraphs according to this comment.

18. Line 343, 'lower' to 'lowest'.

Thanks for your comment. We will correct it in revised manuscript.

19. Section 4.1.3 should be merged into section 4.1.2.

Thanks for your comment. We will reorganize the structure according to the the comment and other comments about the structure.

20.In figure 4, it should include the obs pattern, or the difference between opt/default and observation.

Thanks for your comment. We will add new results figures in revised manuscript according to this comment.

21.Line 366, it is confusing for this sentence "Therefore, we need to further ..."

Thanks for your comment. In previous sentences. We illustrate that the result of global-based tuning is not significant, some regions still need to be tuned. So that we try to find the best way to use the proposed method to improve the simulation result. Perhaps our choice of words was not precise. We will use more accurate term in this sentence.

**References**

[1] X. Wu, L. Hu, L. Wang, H. Lu, and J. Zheng. Surrogate model-based precipitation tuning for cam5. *Geoscientific Model Development Discussions*, 2023:1–30, 2023.

[2] Jakub Kudela and Radomil Matousek. Recent advances and applications of surrogate models for finite element method computations: a review. *Soft Computing*, 26(24):13709–13733, December 2022.

[3] Timothy W Simpson, Andrew J Booker, Dipankar Ghosh, Anthony A Giunta, Patrick N Koch, and R-J Yang. Approximation methods in multidisciplinary analysis and optimization: a panel discussion. *Structural and multidisciplinary optimization*, 27:302–313, 2004.

[4] Rommel G. Regis and Christine A. Shoemaker. A Stochastic Radial Basis Function Method for the Global Optimization of Expensive Functions. *INFORMS Journal on Computing*, 19(4):497–509, November 2007.

[5] Ky Khac Vu, Claudia d'Ambrosio, Youssef Hamadi, and Leo Liberti. Surrogate-based methods for black-box optimization. *International Transactions in Operational Research*, 24(3):393–424, 2017.

[6] Yong Wang, Da-Qing Yin, Shengxiang Yang, and Guangyong Sun. Global and local surrogate-assisted differential evolution for expensive constrained optimization problems with inequality constraints. *IEEE transactions on cybernetics*, 49(5):1642–1656, 2018.

[7] Hugh Morrison and Andrew Gettelman. A new two-moment bulk stratiform cloud microphysics scheme in the community atmosphere model, version 3 (cam3). part i: Description and numerical tests. *Journal of Climate*, 21(15):3642–3659, 2008.

[8] Andrew Gettelman, Hugh Morrison, and Steven J Ghan. A new two-moment bulk stratiform cloud microphysics scheme in the community atmosphere model, version 3 (cam3). part ii: Single-column and global results. *Journal of Climate*, 21(15):3660–3679, 2008.

[9] Andrew Gettelman, Xiaohong Liu, Steven J Ghan, Hugh Morrison, Sungsu Park, AJ Conley, Stephen A Klein, James Boyle, DL Mitchell, and J-LF Li. Global simulations of ice nucleation and ice supersaturation with an improved cloud scheme in the community atmosphere model. *Journal of Geophysical Research: Atmospheres*, 115(D18), 2010.

[10] GJ Zhang and NA McFarlane. Sensitivity of climate simulations to the parameterization of cumulus convection in the canadian climate centre general circulation model. *Atmosphere Ocean*, 33:407–446, 1995.

[11] Richard B Neale, Jadwiga H Richter, and Markus Jochum. The impact of convection on enso: From a delayed oscillator to a series of events. *Journal of climate*, 21(22):5904–5924, 2008.

[12] S Park, CS Bretherton, and PJ Rasch. The revised cloud macrophysics in the community atmosphere model. *Journal of Climate*, 2010.

[13] Yun Qian, Huiping Yan, Zhangshuan Hou, Gardar Johannesson, Stephen Klein, Donald Lucas, Richard Neale, Philip Rasch, Laura Swiler, John Tannahill, Hailong Wang, Minghuai Wang, and Chun Zhao. Parametric sensitivity analysis of precipitation at global and local scales in the Community Atmosphere Model CAM5. *Journal of Advances in Modeling Earth Systems*, 7(2):382–411, June 2015.

[14] Stefan M Wild, Rommel G Regis, and Christine A Shoemaker. Orbit: Optimization by radial basis function interpolation in trust-regions. *SIAM Journal on Scientific Computing*, 30(6):3197–3219, 2008.

[15] Ruobing Chen, Matt Menickelly, and Katya Scheinberg. Stochastic optimization using a trust-region method and random models. *Mathematical Programming*, 169:447–487, 2018.

[16] Kely DV Villacorta, Paulo R Oliveira, and Antoine Soubeyran. A trust-region method for unconstrained multiobjective problems with applications in satisficing processes. *Journal of Optimization Theory and Applications*, 160:865–889, 2014.

[17] Xin-Zhong Liang, Hyan I Choi, Kenneth E Kunkel, Yongjiu Dai, Everette Joseph, Julian XL Wang, and Praveen Kumar. Development of the regional climate-weather research and forecasting (cwrf) model: surface boundary conditions. *Scientific report*, 1, 2005.

[18] Richard B Neale, Chih-Chieh Chen, Andrew Gettelman, Peter H Lauritzen, Sungsu Park, David L Williamson, Andrew J Conley, Rolando Garcia, Doug Kinnison, Jean-Francois Lamarque, et al. Description of the ncar community atmosphere model (cam 5.0). *NCAR Tech. Note NCAR/TN-486+ STR*, 1(1):1–12, 2010.

---

## Author Comment (AC4)

**Replies to Referee #2, GMD-2023-164**

Xianwei Wu, Liang Hu, Lanning Wang, Haitian Lu, and Juepeng Zheng

November 30, 2023

Thank you very much for your patient and detailed comments on our work [1]. These valuable comments are very helpful for us to improve this paper. After carefully reading all the questions, we have answered each of them and will make appropriate corrections in the revised version of our manuscript.

In this attachment, the blue paragraphs represent your comments, and the black paragraphs below are our corresponding replies.

**1    Replies to 1-5 questions**

Replies to these questions are as follows:

1. The presentation of the algorithm and techniques could be more concise. It would further benefit from clear mathematical notation and equations, a clear nomenclature, and a clearer order. For instance, RMSE is used before properly introduced. The calculation of V$\hat{s}$ and V$\hat{d}$ are hard to follow and not right away clear.

Thanks for your comment. CAND (candidate points) strategy is proposed in [2], has been widely used in surrogate model-based optimization method [3, 4]. It is used for balance the global exploration and local exploitation. **Please note that the terms "global and local" represent characters of the optimal solution in optimization methods rather than "best parameter for global precipitation or region precipitation".** The inputs of the strategy are a set of generated points $\Omega$, a set of initial sampling points $A$, and the output of the strategy is the best candidate point selected from the generated points $\Omega$.

In each iteration step, there are two criteria in CAND to select to select the best candidate point:

1). Estimated function value obtained from the surrogate model.

2). Minimum distance from previously evaluated points.

The first criterion represents the exploitation, which means that search a better solution based on known regions. The second criterion represents the exploration, which means that search a better solution in an unknown region. **Please note that in this sentence "region" represents a part of parameter space rather than precipitation simulation results over each "region" like East Asia.** In order to find the next candidate for evaluation, we do not minimize that fitness function over a continuous set. Instead, we select the best among a finite set of randomly generated points. Fitness functions are made up from these two criteria: the value of fitness function obtained from the surrogate model at each point and its minimum distance to existing data points.

The description of CAND is described in Algorithm 1:

Where, the $\Omega$ represents the random samples generated in this iteration process. $S(x)$ represents the predict value of point $x$ generated by surrogate model. $\Delta(x)$ is the minimum distance from point $x$ to the current sampling point set $A$ and $y$ represents each point in set $A$. In line 1, fitness function values of generated point sets are calculated according to the surrogate model and their maximum
* * *
**Algorithm 1** Candidate point strategy
* * *
1: Compute $s^{max} \leftarrow \max_{x \in \Omega} s(x)$ and $s^{min} \leftarrow \min_{x \in \Omega} s(s)$

2: **for** each $x \in \Omega$ **do**

3: $\quad V^S(x) = \begin{cases} \frac{s(x) - s^{min}}{s^{max} - s^{min}} & if \ s^{max} > s^{min} \\ 1 & else \end{cases}$

4: $\quad$ Calculate corresponding value of objective function for each sample.

5: **end for**

6: **for** each $x \in \Omega$ **do**

7: $\quad \Delta(x) = min_{y \in A} d(x, y);$

8: **end for**

9: Compute $\Delta^{max} \leftarrow \max_{x \in \Omega} s(x)$ and $\Delta^{min} \leftarrow \min_{x \in \Omega} s(x)$

10: **for** each $x \in \Omega$ **do**

11: $\quad V^D(x) = \begin{cases} \frac{\Delta(x) - \Delta^{min}}{\Delta^{max} - \Delta^{min}} & if \ \Delta^{max} > \Delta^{min} \\ 1 & else \end{cases}$

12: **end for**

13: **return** $argmin_{x \in \Omega} w V^S(x) + (1 - w) V^D(x)$
* * *
and minimum are marked. In line 2-4, the value of $V^S$ is calculated and the for loops represents the criterion 1, a smaller value of $V^S$ means that the current point is an effective exploitation. In line 6-8, the minimum distance $\Delta(x)$ between the point sample $\Omega$ and current sampling point set is calculated in the for loops. The maximum and minimum of $\Delta(x)$ are marked in line 9. The for loops in line 10-12 represent the criterion 2, a smaller value of $V^D$ means that the current point is an effective exploration. The last line means the weighted sum to balance the exploration and exploitation.

We will add these to the revised manuscript in section 3.2.3.

We will improve the structure of the study in revised manuscript.

2. The paper lacks describing links between the physics and the choice of parameters. Why do certain parameter combinations perform better (for instance, what do they affect, how does that affect the general performance etc.)

Thanks for your comment. We try to reply to this comment from the following points.

1).The purpose of this work is to improve the CAM5 precipitation simulation result accord to parameter tuning method, rather than analyzing the mechanism of physics process. We believe that the goal has been achieved and it is a complete work. In general, the calibration of parameters in parameterization schemes relies on statistical models and expert knowledge, leading to significant uncertainty. Small variations in parameter values can result in substantial changes in simulation results. In this paper, we propose a surrogate model based method which can quickly calibrate parameters, and improve CAM5 precipitation using the proposed multi-level surrogate model method. During the optimization process, we found that the same parameter value has different effects on simulation results in different regions, called "rocker effect". This suggests that it is challenging to achieve precipitation optimization for all regions using the same set of parameters. Therefore, we design a non-uniform parameterization scheme, employing different parameter values for distinct regions. We find a more suitable set of parameters for each region according to the multi-level surrogate model-based method and integrate these different sets of parameter values into one case.

2).We do not change the physical process in the parameterization schemes, we only changed the values of the parameters, or different values in different regions. In this paper, we use CAM5 and the selected parameters belong to three different parameterization schemes: The cloud microphysics parameterization scheme is proposed by [5, 6, 7]. The deep convective parameterization scheme is proposed by [8] and modified by [9]. The cloud fraction parameterization scheme is proposed by [10].

We believe that these studies have analyzed the physical meanings of each parameter and their effects on simulation results. While maintaining these physical processes, we improve the parameterization scheme based on the optimal values obtained for each region through the proposed tuning methods. In the improved parameterization scheme, the parameter values for each selected region are set to the tuned values, while the values for other regions remain at default parameters. In the parameterization scheme file, there have been no changes to the descriptions of physical processes. The modifications we made involve selecting different numerical values based on the judgment of region latitudes and longitudes. So that we believe that these positive improvements achieved by changing the values of the parameters. In [11], there is more introduction to the mechanism of physics process, including the impact of each parameter on precipitation results in different regions. We also refer to this work when selecting parameters and determining the range of these parameters. We will try to explain these effects from the parameter value changes according to this work and any other related works.

3. The paper does not discuss that in general tuning for a single metric is not required as a climate model has many different metrics that need to be fulfilled. Thus, it is required to discuss how the precipitation tuning might degrade other fields. For instance, what is the effect on the global mean temperature from this tuning etc.

Thank four your comment. We try to reply to this comment from the following points.

1) In this paper, we propose a surrogate model-based parameter tuning method to improve CAM5 precipitation simulation results. The proposed method belongs to a single objective optimization method. Throughout the entire optimization process, all methods, strategies, and objective functions are aimed at reducing the Root Mean Square Error (RMSE) between CAM5 simulated precipitation and reanalysis data. Precipitation is a key physical process linking many aspects of climate, weather, and the hydrological cycle, and changes in precipitation regimes and characteristics are of great importance to energy, society, and ecosystems[11]. So that in this paper, precipitation is selected as the target for analysis and research. The final experimental results demonstrate that our approach can improve the simulation of CAM5 precipitation. This method maybe cannot optimize multiple objectives simultaneously. Multi-objective surrogate model-based parameter tuning method is also one of our future research topics. Therefore, in this paper, for the analysis of the optimization results, we only consider tuning the parameter values to change the simulation results of precipitation.

2) In our experiments, the selected compset is F_2000_CAM5, and its sea surface temperature (SST) input data is in the form of climatology, which is fixed. If we use the CMIP case with variable SST data, we may observe more changes in temperature or other simulation results. We believe that in this situation, it is possible to more significantly observe changes in other metrics. By using different compsets , we aim to study more variables, and combine with the multi-objective parameter tuning methods mentioned earlier, it is one of our future research plans.

3)Although we propose a single-objective parameter tuning method, and select the compset with a climatological sea surface temperature (SST) as input data, we will attempt to discuss, in the simulation results of non-uniform parameterization schemes, the impact of parameter changes on indicators other than precipitation, such as temperature, pressure and specific humidity. Perhaps these changes are small or not positive. The discussion regarding these metrics will be added to section 4.4 in the revised manuscript.

4. The presentation of introducing the non-uniform parameter values is not entirely clear. Why should that be? What is the physical explanation for using different parameter values in different places? Shouldn't the physics be independent of the location particularly in regions which are relatively similar (South Pacific, Nino?)? Particularly you tune for different ocean regions; it is not clear why

different ocean areas should have different parameter tunings. (I could understand a land vs ocean parameter change, however, different oceans or land masses requires more careful introduction and physical justification)

Thanks for your comment. We try to reply to this comment from the following points.

1) In this paper, the motivation for the nonuniform parameter parameterization scheme is as follows:

- It is well known that CAM5 is a well tuned model, however the holistic optimal parameters do not necessarily mean they are the best solutions for every region. The simulation results over these regions are challenging to improve through global tuning experiments. **Please note that in this sentence "global" tuning represents the CAM5 tuning experiments to improve the global simulation results of precipitation rather than "global optimal" in optimization process.**

- Regional optimization experiments demonstrate that some regions have optimal parameters, leading to better results than default parameters. However, applying these parameters to global simulations may not obtain optimal results; they are effective only within the selected regions for achieving the best simulation outcomes.

- Our experiments show that there is a "rocker effect" in the influence of parameters on precipitation. The same parameter values have different effects on different regions. When the simulation results in one region improve due to changes in parameter values, the results in other regions may decline. This implies that optimizing precipitation for all regions using a single set of parameters is challenging.

2) In [11], the authors discuss the contributions of different parameters to precipitation in different regions. The research results indicate that the contribution of different parameters to precipitation varies across regions. As shown in the figure 1, it can be observed that the contribution of parameters to precipitation cannot be simply judged based on the relationship with ocean or land. Even in adjacent regions, there can be some degree of differences. When using globally uniform parameter values, in order to pursue a holistic optimal solution, approximate mean value is employed to achieve a better overall simulation performance. If there is a significant difference between the local optimum and mean value in certain regions, the simulation results for that region will have a large error. **Please note that "local optimum" in this sentence means that the best parameters over this region, rather than "Local optimum solution" in the optimization process.**

3) We must also consider the diversity of the oceans regions. In CAM5, the physical processes related to the ocean include optical reflection and some complex thermodynamic processes. The variations in sea surface temperatures have a significant impact on these physical processes, leading to substantial differences in simulation results across different regions. As can be seen in the Figure 2, the sea surface temperatures vary significantly across different oceanic regions, such as the Pacific region and warmpool. In the presence of such differences, dividing parameters based solely on ocean/land distinctions is not precise enough. Therefore, we try to select multiple regions and utilize faster parameter tuning method to find better parameters. These papameters are then integrated into the same case through a non-uniform parameterization scheme.

In summary, we proposed the nonuniform parameter parameterization scheme. We search different parameter combinations for different regions by surrogate model based tuning method and integrate them into a single case in a non-uniform parameterization scheme.

5. The presentation is unclear as to why is a GP only used for the regional-level surrogate models and not for the global?

[Figure]

Figure 1: Global spatial distributions of relative contributions (%) of parameters to total variance of annual mean precipitation in [11].

[Figure]

Figure 2: Climatology SST

[Figure]

Figure 3: Local surrogate model cross-validation results

Thanks four your comment.

In the process of constructing the global surrogate, because of the relatively large amount of samples, we chose some relatively complex learning-based models and selected the optimal method based on cross-validation results. The results in this paper indicates that the GBRT is the best method to construct global surrogate model.

In the process of building local surrogate models, we take into account the insufficient number of samples. Using some relatively complex learning-based methods may lead to under-fitting. Therefore, we consider selecting some relatively simple construction methods. Among them, Polynomial Response Surface (PRS), Gaussian Process (GP), and Radial Basis Function (RBF) are the most commonly used regression-based or statistics-based methods, widely used for various complex parameter optimization problems in the industry [12]. Therefore, we consider choosing one of these three as the method for constructing local surrogate models. Among them, we first consider PRS. Although it is the simplest method, the fitting performance of the PRS model to complex curves is relatively poor. PRS model is a simple model based on polynomials, and it may not perform well for complex, nonlinear, or highly interactive systems. Its expressive capacity is limited and may not accurately capture certain complex relationships. According to [13], the polynomial surrogate model does not perform well in terms of fitting accuracy for multivariate and nonlinear problems. Therefore, we choose either GP or RBF to construct the local surrogate model.

In order to choose a model that is more suitable for our study, we conducted cross-validation experiments based on the selection method for the global model. We selected three learning-based methods: Random Forest (RF), Support Vector Machine (SVM), and Artificial Neural Network (ANN) for comparison. The results are shown in the figure 3. The results indicate that compared with RBF method, GP has a smaller error in cross-validation, providing more accurate predictions. Moreover, the cross-validation results are better than the three learning-based methods. In contrast, the RBF method not only has a larger error but also a wider range of upper and lower relative error bounds. The prediction results are unstable, and the predictive performance is lower than the three learning-based surrogate model construction methods.

Please note that in these paragraphs, "global" and "local" represent the different surrogate model, rather than "global/local optimal" in optmization process or simulation results in "global/region".

We will add the relevant experimental results to the revised manuscript.

**2  Replies to major comments**

Replies to major comments are as follows:

1. LL.37-70: it would be worth discussing also the tuning approaches by Hourdin and Williamson in more detail

(http://link.springer.com/10.1007/s00382-013-1896-4 ;

http://link.springer.com/10.1007/s00382-014-2378-z ;

https://gmd.copernicus.org/articles/10/1789/2017/ ;

https://agupubs.onlinelibrary.wiley.com/doi/10.1029/2020MS002423 ;

https://onlinelibrary.wiley.com/doi/10.1029/2020MS002225 ;

https://onlinelibrary.wiley.com/doi/10.1029/2020MS002217 )

Thank you for your suggestion. These works have many highlights in the description of the methods and the explanation of the physical mechanisms. For example, In [14, 15], authors discuss machine learning for ESM calibration and use machine learning method on both single-column model and global model. In [16, 17], authors propose a history matching method to analyse the uncertainty of parameters in HadCM3. The author also proposed the concept of "over-tuning" to prevent over fitting in the tuning results.

We believe that our work is also an extension of these tasks and a supplement to certain aspects. They are very helpful in improving our manuscript.

2.Ll.135-138: How do you inform the parameter range of the parameters to tune for? How do you choose exactly those parameters? Generally, there are more parameters in the parameterizations; why exactly those 6?

Thanks for your comment. We know that the sensitivity of parameters has an important impact on pattern tuning. If insensitive parameters are selected for disturbance and adjustment, the simulation results will hardly change significantly. Therefore, the selection of sensitive parameters is one of the important conditions for CAM5 parameter tuning. Many previous studies have conducted sensitivity-related studies on precipitation-related parameters in CAM5. The parameter range and the reason we choose these parameters are according to these studies [11, 18]. These parameters we selected are proved most to sensitive precipitation in these studies. Based on these studies, we calibrate the parameters sensitive to precipitation determined in these research works. We propose a surrogate model-based approach, and the results show that our method can improve the simulation performance of precipitation in CAM5.

3. LL. 140-141: Why do you choose particularly those regions? Why are they important?

Thanks for your comment. We try to reply to this comment from the following points.

1) These regions and there range are selected according to [19]. In this study, authors discuss the characteristics over these regions. We select these regions and determine their ranges based on this study.

2) These regions have a high amount of precipitation value. These regions are distributed in the $45°N - 45°S$. As can be seen in the middle image in Figrue 4, The majority of global precipitation is concentrated in the mid-low latitude regions. tuning parameters over these regions can effectively

improve the simulation of CAM5 precipitation.

3) As can be seen in Figure 4, in default experiment, these regions have a certain degree of error compared to observational data. Since CAM5 is a well-calibrated model. Our experiment also proves that the global-oriented tuning effect is not significant. Despite the overall good simulation results, there are still errors in these regions. To improve the simulation results in these areas, new methods need to be explored.So that we try to research these regions, find a set of parameters to improve the simulation results over these regions by the proposed surrogate model-based method.

4. Section 3: I would suggest sticking to terminology! Whenever you talk about an actual model simulation I would suggest using ESM/CAM5 or something like that. Otherwise you start mixing up terms such as global-model, global-level, complex model which does not make it easy to follow which of all the models you refer to or whether it is a new one.

Thanks for your comment. In this paper "global-model" and "global-level" model are equivalent. They represent the global-level surrogate model in the proposed method. "Complex model" represents the optimization problem which the fitness function is hard to calculate. Perhaps the word "model" has different meanings is different terms such as "surrogate model", "complex model" and "earth system model". In order to avoid ambiguity, "complex model" can be replaced as "optimization problem", "global-level model","global model" can be replaced as "global surrogate model". "local model", "local-level model" can be replaced as "local surrogate model". We will standardize the use of these terms in revised manuscript.

5. L 213: Why do you use reanalysis data? Why not GPCP for instance?

Thanks for your comment. Both ERA5 and GPCP can be used for precipitation analysis. They both provide global precipitation data. However ERA5 provides higher-resolution data. Data of GPCP are provided on a 2.5 degree grid and ERA5 precipitation data are provided on a 0.25 degree grid. We believe that choosing data with higher resolution can significantly contrast the tuning results, thereby demonstrating the effectiveness of the proposed method. So that we select ERA5 instead of GPCP as the metric for precipitation parameter tuning. The RMSE is calculated between the CAM5 simulation results and ERA5 reanalysis data.

6. Equation 1: Comparing CAM simulations with reanalysis requires regridding data. What technique do you use for that?

Thanks for your comment. The CAM5 case we used in this paper is spectral element dynamical core (SE-dycore) formulation. It need be regridded to compare with reanalysis data. We know that the ERA5 reanalysis data is lat/lon grid. We regrid the simulation result to lat/lon grid. In this paper, NCL language is utilized for data processing, calculations, and visualization. This choice is made because NCL exhibits strong capabilities in handling CESM output data, and its operations for reading data and creating visualizations are straightforward. So that we use NCL to regrid CAM simulation data to lat/lon grid. Regridding function "ESMF_regrid" is select to complete the regrid operation and "bilinear", which is also the default parameter of the function "ESMF_regrid", is used for the regridding interpolation method as the function input parameter.

7. L. 249: What is the level of the fitness function? Equations for Vŝ and Vd̂ would be very helpful!

Thanks for your comment. "The level of the fitness function" means the quality of the samples, in this paper, it means the fitness values (RMSE values) of these samples. The process of CAND is shown in Section 1 question 1. It can be seen that $V^s$ represents the exploitation mechanism, it describes the simulation result of the sample $\Omega$ obtained by the current surrogate model. $V^d$ represents the exploration mechanism, it expresses the exploration for the unknown region of the current surrogate model. $V^s$ and $V^d$ is used to balance exploitation and exploration. We will add the description of

[Figure]

Figure 4: The precipitation distribution of default experiment, there are default experiment, observation data, difference between default experiment and observation data from top to bottom.

CAND to the revised manuscript.

8. L. 267: "..., allowing us to estimate uncertainty from the weight parameter": How do you estimate uncertainty of the weight parameter?

Thanks for your comment. Perhaps there is some ambiguity in our expression, in this sentence, the weight parameter represents $V^D$ in CAND strategy, we use the parameter $V^D$ to represent the exploration mechanism, which means searching a new optimum in uncharted regions.

The sentence does not means that we want to estimate the uncertainty of the parameters on the CAM5 simulation results.

Maybe these terms like "uncertainty", "weight parameter" have other meanings in ESM parameter tuning. We revise the sentence as follows:

**The set of the previously sampled points denote the region which has been explored, allowing us to estimate the uncertainty from the distance between the generated point sets to the explored region.**

9. Ll. 222-225: The two sentences appear to have very similar information and should be rewritten

Thanks for your comment. The two sentences both describe that add new samples to update surrogate model. We delete the sentences with similar meanings. The new sentences are as follows:

**Generally, when solving a complex parameter optimization problem by a surrogate model, to improve the accuracy simulation results of the surrogate model, additional new sample points need to be added. Thus reducing the number of simulations of the actual complex model.**

We will add these sentences in revised manuscript.

10. Figure 1: Could you put the whole algorithm into the flowchart which indicates which technique is used at which step which would make the whole description much clearer.

Thanks for your comment. We will update the flowchart according to your comment, in each step we will add which technique is used. The new flowchart are shown in Figure 5. We believe that the new flowchart is much clearer. Compared with the old flowchart, we add the whole process of CAND and trust region method. In Figure 5. We add an schematic diagram for updating of the surrogate model during the iteration, providing a clearer illustration of each iteration and incorporating each step of CAND and the trust region in different optimization phases. This enhancement allows for a more lucid representation of the iterative process."In order to show the integration of these process, dotted bordered rectangle is used to mark the method which the current process belongs to. In addition, we add the surrogate construction method of each level in the new flowchart.

We will add the new figure in revised manuscript.

11. Figure 3: Why is the relative error of the proposed method increasing with iterations? Shouldn't the surrogate model get more accurate with iteration numbers?

Thanks for your comment. Ideally, the error will gradually decrease, but there will be some inevitable small oscillations during the optimization process, resulting in a slight increase in error.

it's possible that our method may indeed have slightly higher errors compared to ASMO in the end. However, our method demonstrates greater stability throughout the entire optimization process, with errors consistently maintained at a lower level. In contrast, AMSO exhibits initial oscillations in errors, indicating that our surrogate model remains stable. While our final error may be slightly higher than that of ASMO, we believe that in cases where the errors are relatively close, the reduction in the number of optimization iterations is a highlight of our method.

12. L 256: "... we only run the real model once, and": But don't you add more than one sample in each iteration? So, how do you need ot run the model only once?

[Figure]

Figure 5: New flowchart

Thanks for your comment. We run the real model (CAM) once in each iteration step, and add the parameters and corresponding RMSE value to sampling set. For example, if the whole tuning process contains 30 steps, in each step one pair of parameters and RMSE value are added to sampling set. There are 30 pairs added to sampling set. We can rewrite the sentence as:

**For the whole tuning process, we only run the CAM5 model once in each iteration step, and all of the samples are predicted using the surrogate model.**

We believe that new sentences will express their meanings more clearly, and we will add them in revised manuscript.

13. LL. 377-384: "I think it is important to understand how each parameter influences the model simulations. Why did you pick only one here? This section requires more careful exploration of the parameter itself and the physical mechanisms. What are the physical reasons for the positive and negative correlations described in ll. 377-379

Thanks for your comment. We try to reply to this comment from the following points. 1) We agree your comment that it is important to understand how each parameter influences the model simulations. The motivation of this work is to propose a tuning method and how to use the mothod for CAM5 precipitation tuning. We analyse the parameter rhminl to prove the "rocker effect": The values of parameters will have different impacts in different regions. Thus introduce a more appropriate way to use the surrogate-based method and the nonuniform parameter parameterization scheme. The core of this work is parameter tuning. We aim to find parameter combinations that improve the precipitation simulation results of CAM5. The surrogate modeling is a rapid approach to identify optimal parameters. The non-uniform parameterization scheme involves integrating optimal parameters for different

Table 1: The CAM simulation performance increase for each region.

| Region | Default RMSE | Global surrogate RMSE | Optimized RMSE | Reduction Rate |
|---|---|---|---|---|
| WarmPool | 1.985 | 1.961 | 1.924 | 3.07% |
| South Pacific | 0.855 | 0.788 | 0.455 | 46.78% |
| Niño | 0.931 | 0.855 | 0.773 | 17.04% |
| South America | 2.576 | 2.459 | 2.371 | 7.94% |
| South Asia | 1.484 | 1.352 | 1.293 | 12.87% |
| East Asia | 1.213 | 1.043 | 0.878 | 27.68% |

regions. The analysis of the "rocker effect" is conducted merely to demonstrate the existence of different optimal parameters in different regions. We are willing to study how each parameter influences the model simulations in future research, which contributes to propose the more efficient tuning methods.

2)We believe that this phenomenon is attributed to the design of the physical processes in the parameterization scheme and the selection of default parameters. Since we don't alter the physical processes in the parameterization scheme, it is likely that this phenomenon is intrinsic to the parameterized physical processes. Alternatively, it could be due to the setting of default parameter values, as mentioned earlier. Default parameter values are based on statistically obtained means. Consequently, when perturbing parameter values around the default settings, simulation results may exhibit different trends in various regions.

14. Table3/4: Could you also discuss the RMSE of those regions from the optimized parameter set from the global-surrogate model? This would clarify what the gain is from the local-level surrogate models.

Thank four your comment. We agree your comment, discuss the improve from different level of surrogate model will better demonstrate the effectiveness of our method. We will add the tuning result obtained from global surrogate model. The new table are shown in Table 1.

where, the "Global surrogate RMSE" means the optimization result obtained by global surrogate model and "Optimized RMSE" represents the final result.

Table 4 in manuscript represents the results of nonuniform parameter parameterization scheme, which are obtained by the CAM5 simulation integrated optimal parameters over each regions. These data are derived from the simulation results that integrate parameters from different regions, without involving any surrogate models or new optimization processes.

**Please note that in these sentences, "global" and "local" represent the different surrogate model, rather than "global/local optimal" in optmization process or simulation results in "global/region".**

**3   Replies to minor comments**

Replies to minor comments are as follows:

1. L. 1: "The uncertainty of physical parameters is a major reason for a poor precipitation simulation performance in Earth system models (ESMs), especially over the tropical and Pacific regions.": Is it not only uncertainty of physical parameters but also the microphysics parameterizations itself.

Thanks for your comment. We agree your comment, microphysics parameterization is also one of the main factors influencing the precipitation. Our wording might be too absolute. We should use words like "one of the major reasons".

We can rewrite the sentence as:

**The uncertainty of physical parameters is one of the major reason for a poor pre-**

**cipitation simulation performance in Earth system models (ESMs), especially over the tropical and Pacific regions.**

2. Ll. 14-15: "The results show that the surrogate model-based optimization method can significantly improve the simulation performance of the CAM model.": I would rephrase it stating that the surrogate model-based optimization method allows for better identifying optimal parameter values.

Thanks for your comment. We agree your comment, we will revise the sentence according to your comment.

We can rewrite the sentence as:

**The results show that the surrogate model-based optimization method can allow for better identifying optimal parameter values of CAM5.**

3. L. 25: "...could lead to huge deviations in the simulations": Deviations from what?

Thanks for your comment. We know that some parameters are sensitive for the simulation result value (eg. temperature, precipitation). A slight change in these parameter values can lead to significant numerical variations in the simulation results. Perhaps our choice of words was not precise. "Error" might be a more accurate term in this sentence.

We can rewrite the sentence as:

**These parameters control the physical processes at the subgrid scale, and slight variations could lead to huge errors in the simulations.**

4. LL. 168-169: "The strategy leverages the information and knowledge obtained from the surrogate model to optimize the run time of the real complex model to fulfill the requirement of accuracy.": How do you optimize for the run time of the real complex model (I guess the ESM)?

Thanks for your comment. The real complex model is the ESM in this paper, we will revise these terms to avoid ambiguity. The appropriate strategy can reduce the number of iteration so as to reduce run time of the real complex model. For traditional parameter optimization algorithms such as Genetic Algorithm (GA) and Particle Swarm Optimization (PSO), one optimization iteration may require evaluating the objective function several hundred times. In our case, this means running CAM5 simulations hundreds of times, and the computational cost this is unacceptable. In the surrogate model-based method, although we also evaluate the objective function several hundred times, these evaluations are based on the surrogate model. Only the selected set of parameters will be sent to the CAM5 simulation to update the surrogate model. Therefore, our method reduces significant computational resource expenses. Perhaps our choice of words was not precise. We will use more accurate term in this sentence.

The sentence can be revised as:

**This strategy leverages information and knowledge obtained from the surrogate model to reduce the number of runs of the CAM5, meeting the requirement for accuracy.**

5. L 171: ".. to update the global-level surrogate model until global-model convergence.": Which global-model do you refer to here, which global-model has to converge?

Thanks for your comment. The global model is the global-level surrogate model. Only when the global-level surrogate model phase converges, the local-level surrogate model will be created and the method. The proposed surrogate model-based tuning method is divided into two optimization phases. The first phase is based on global surrogate model. The second phase is local surrogate model search phase. When the first phase converges, the second optimization phase begins. **In this sentence, the "global/local" represents the global/local-level surrogate model rather than CAM5 simulation results in global/region.** We will revise these terms to avoid ambiguity.

The sentence can be revised as:

**The recently generated sample points and their corresponding simulation outputs are added to the initial sample sets and are used to update the global-level surrogate model until global surrogate model phase convergence.**

6. L.172: "... high-waulity CAM" Do you mwan with high-waulity simulations closer to the target value?

Thanks for your comment. "The high-quality results" means the simulations closer to the target value, in this paper, is the reanalysis data.

7. L. 175-176: "In the parameter tuning process, each surrogate model can fully explore the parameter space to obtain better solutions, generating a large number of samples.": I don't understand this sentence as earlier (ll 171-172) it is stated that local-level surrogates don't use the whole parameter space?

Thanks for your comment. "Parameter space" means the range of all the parameters, local-level surrogate don't use the whole samples, only high quality samples selected. However the parameter searching in tuning process of local-level surrogate is also over the range of all the parameters (parameter space). Throughout the entire optimization process, in any optimization phase, we do not change the range of parameters; they still adhere to the ranges described in Table 2 in manuscript.

8. LL. 202-211: Why do you talk about 1-D LHS. Usually LHS code can handle several dimensions. LHS code usually makes sure to maximize the minimal distance between all vectors in order to sample the whole space as uniformly as possible.

Thanks for your comment. We agree your comment that LHS code can handle several dimensions. We use a LHS to generate samples, there are 6 parameters selected in this paper. Introduction of 1-D LHS is a example to describe the process of LHS. We use 6-d LHS in this paper. We believe that the 1-D LHS process is more intuitive, making it easier to understand the principles and procedures of the LHS method.

9. Ll. 222-225: The two sentences appear to have very similar information and should be rewritten.

Thanks for your comment. We will rewrite these sentences in revised manuscript.

The new sentences are as follows:

**Generally, when solving a complex parameter optimization problem by a surrogate model, to improve the accuracy simulation results of the surrogate model, additional new sample points need to be added. Thus reducing the number of simulations of the actual complex model.**

10. ll. : 347: Do you compare here the RMSEs of the final optimized parameter set? If so, do they converge to the same parameter set or different ones?

Yes, we compare RMSE of the simulation results of different parameters. However, they converge to different parameter values.

11. Figure 4: It would be good to see what you are tuning for. Can you also add the target value and not only the default and the biases to the target?

Thanks for your comment. We will add these figures according to your comment.

12. L 370: What do you mean with influence mode?

Thanks for your comment. The "influence mode" means the impact of changes in parameter values on the precipitation of each grid, positive correlation or negative correlation. In the experiment, when we change the parameter values, the precipitation in different grid points showed varying responses. Some regions might improve, while others might worsen. It's challenging to use the same set of parameters to enhance precipitation in all regions. Therefore, we propose a non-uniform parameterization scheme, attempting to use different parameter combinations in different regions.

**4   Replies for technical issues**

Replies for technical issues are as follows:

1. Labels on contour plots are generally very small.

Thanks for your comment. We will revise these plots according to this comment.

2. L. 63: I might have missed it but "ANNs" acronym was not introduced.

Thanks for your comment. Acronyms will be properly defined in our revised manuscript.

3. L. 78: I might have missed it but "SCA-SMA" acronym was not introduced.

Thanks for your comment. Acronyms will be properly defined in our revised manuscript.

4. L.132: "The compset used in this study is F_2000_CAM5, and the resolution is ne30_g16": To the normal reader these abbreviations don't mean anything. A little bit more explanation would be nice. What is F_2000_CAM5 for instance or what does ne30_g16 mean in the physical world?

Thanks for your comment. We will add more description about CAM5 and the compset used in this study. They include the modes used in the compset, the description of the grid and their specific meanings. We will supplement to the revised manuscript based on [20].

5. Can you use maybe mathematical notation of the original parameters in the paraemeterization instead of the CAM5 parameter naming? For instance line. 148 zmconv_tau is simply tau in the original Zhang McFarlane paper.

Thanks for your comment. We will use mathematical notation of the parameters in revised manuscript.

6. L 275: "actual situation": you probably mean behaviour or something like that.

Thanks for your comment. We agree your comment, it may be more accurate if we use "behaviour" to replace to the "situation".

7. Figure 2: y-axis label missing

Thanks for your comment. We will correct it in revised manuscript.

8. L. 333: What is now X,Y in the S{X,Y} notation?

Thanks for your comment. {X,Y} represents the pairs of parameters and corresponding RMSE value. X represents the parameter combinations of selected 6 parameters and Y represents the RMSE values obtained from CAM simulation results of corresponding parameters.

9. Figure 3: no x-axis label

Thanks for your comment. We will correct it in revised manuscript.

10. L. 398: "precipitation change trend": What trend do you mean here? Time trend?

Thanks for your comment. The "trend" means that South Pacific region generally reveals a ladder-like decline from east to west, which is described in last sentence.

**References**

[1] X. Wu, L. Hu, L. Wang, H. Lu, and J. Zheng. Surrogate model-based precipitation tuning for cam5. *Geoscientific Model Development Discussions*, 2023:1–30, 2023.

[2] Rommel G. Regis and Christine A. Shoemaker. A Stochastic Radial Basis Function Method for the Global Optimization of Expensive Functions. *INFORMS Journal on Computing*, 19(4):497–509, November 2007.

[3] Ky Khac Vu, Claudia d'Ambrosio, Youssef Hamadi, and Leo Liberti. Surrogate-based methods for black-box optimization. *International Transactions in Operational Research*, 24(3):393–424, 2017.

[4] Yong Wang, Da-Qing Yin, Shengxiang Yang, and Guangyong Sun. Global and local surrogate-assisted differential evolution for expensive constrained optimization problems with inequality constraints. *IEEE transactions on cybernetics*, 49(5):1642–1656, 2018.

[5] Hugh Morrison and Andrew Gettelman. A new two-moment bulk stratiform cloud microphysics scheme in the community atmosphere model, version 3 (cam3). part i: Description and numerical tests. *Journal of Climate*, 21(15):3642–3659, 2008.

[6] Andrew Gettelman, Hugh Morrison, and Steven J Ghan. A new two-moment bulk stratiform cloud microphysics scheme in the community atmosphere model, version 3 (cam3). part ii: Single-column and global results. *Journal of Climate*, 21(15):3660–3679, 2008.

[7] Andrew Gettelman, Xiaohong Liu, Steven J Ghan, Hugh Morrison, Sungsu Park, AJ Conley, Stephen A Klein, James Boyle, DL Mitchell, and J-LF Li. Global simulations of ice nucleation and ice supersaturation with an improved cloud scheme in the community atmosphere model. *Journal of Geophysical Research: Atmospheres*, 115(D18), 2010.

[8] GJ Zhang and NA McFarlane. Sensitivity of climate simulations to the parameterization of cumulus convection in the canadian climate centre general circulation model. *Atmosphere Ocean*, 33:407–446, 1995.

[9] Richard B Neale, Jadwiga H Richter, and Markus Jochum. The impact of convection on enso: From a delayed oscillator to a series of events. *Journal of climate*, 21(22):5904–5924, 2008.

[10] S Park, CS Bretherton, and PJ Rasch. The revised cloud macrophysics in the community atmosphere model. *Journal of Climate*, 2010.

[11] Yun Qian, Huiping Yan, Zhangshuan Hou, Gardar Johannesson, Stephen Klein, Donald Lucas, Richard Neale, Philip Rasch, Laura Swiler, John Tannahill, Hailong Wang, Minghuai Wang, and Chun Zhao. Parametric sensitivity analysis of precipitation at global and local scales in the Community Atmosphere Model CAM5. *Journal of Advances in Modeling Earth Systems*, 7(2):382–411, June 2015.

[12] Jakub Kudela and Radomil Matousek. Recent advances and applications of surrogate models for finite element method computations: a review. *Soft Computing*, 26(24):13709–13733, December 2022.

[13] Timothy W Simpson, Andrew J Booker, Dipankar Ghosh, Anthony A Giunta, Patrick N Koch, and R-J Yang. Approximation methods in multidisciplinary analysis and optimization: a panel discussion. *Structural and multidisciplinary optimization*, 27:302–313, 2004.

[14] Frédéric Hourdin, Daniel Williamson, Catherine Rio, Fleur Couvreux, Romain Roehrig, Najda Villefranque, Ionela Musat, Laurent Fairhead, F Binta Diallo, and Victoria Volodina. Process-based climate model development harnessing machine learning: Ii. model calibration from single column to global. *Journal of Advances in Modeling Earth Systems*, 13(6):e2020MS002225, 2021.

[15] Fleur Couvreux, Frédéric Hourdin, Daniel Williamson, Romain Roehrig, Victoria Volodina, Najda Villefranque, Catherine Rio, Olivier Audouin, James Salter, Eric Bazile, et al. Process-based climate model development harnessing machine learning: I. a calibration tool for parameterization improvement. *Journal of Advances in Modeling Earth Systems*, 13(3):e2020MS002217, 2021.

[16] Daniel Williamson, Adam T Blaker, Charlotte Hampton, and James Salter. Identifying and removing structural biases in climate models with history matching. *Climate dynamics*, 45:1299–1324, 2015.

[17] Daniel Williamson, Michael Goldstein, Lesley Allison, Adam Blaker, Peter Challenor, Laura Jackson, and Kuniko Yamazaki. History matching for exploring and reducing climate model parameter space using observations and a large perturbed physics ensemble. *Climate dynamics*, 41:1703–1729, 2013.

[18] Raju Pathak, Sandeep Sahany, and Saroj K. Mishra. Uncertainty quantification based cloud parameterization sensitivity analysis in the NCAR community atmosphere model. *Scientific Reports*, 10(1):17499, December 2020.

[19] Ben Yang, Yun Qian, Guang Lin, L. Ruby Leung, Philip J. Rasch, Guang J. Zhang, Sally A. McFarlane, Chun Zhao, Yaocun Zhang, Hailong Wang, Minghuai Wang, and Xiaohong Liu. Uncertainty quantification and parameter tuning in the CAM5 Zhang-McFarlane convection scheme and impact of improved convection on the global circulation and climate. *Journal of Geophysical Research: Atmospheres*, 118(2):395–415, 2013. _eprint: https://agupubs.onlinelibrary.wiley.com/doi/pdf/10.1029/2012JD018213.

[20] Richard B Neale, Chih-Chieh Chen, Andrew Gettelman, Peter H Lauritzen, Sungsu Park, David L Williamson, Andrew J Conley, Rolando Garcia, Doug Kinnison, Jean-Francois Lamarque, et al. Description of the ncar community atmosphere model (cam 5.0). *NCAR Tech. Note NCAR/TN-486+ STR*, 1(1):1–12, 2010.